# Discovery and characterization of gene-by-environment and epistatic genetic effects in a vertebrate model

## Graphical abstract

## Authors

Bettina Welz, Saul Pierotti,
Tomas Fitzgerald, ..., Jakob Gierten,
Joachim Wittbrodt, Ewan Birney

## Correspondence

felix.loosli@kit.edu (F.L.),
jakob.gierten@cos.uni-heidelberg.de
(J.G.),
jochen.wittbrodt@cos.uni-heidelberg.de
(J.W.),
birney@ebi.ac.uk (E.B.)

## In brief

Welz and Pierotti et al. combined high-throughput heart rate phenotyping across temperatures with genome-wide analyses in medaka to identify QTLs and extensive gene-by-environment and epistatic effects, validated through gene editing. Simulations showed that GWAS outcomes depend on model specification. This work highlights the power of controlled model organism studies for dissecting complex trait architecture.

## Highlights

- Medaka used to dissect the genetics of heart rate variation at different temperatures

- 16 QTLs linked to heart rate, with gene-by-environment and epistasis effects resolved

- Gene editing confirmed four candidate genes affecting heart rate across temperatures

- Simulations show how statistical model choice shapes GWAS discovery power

Welz et al., 2026, Cell Genomics 6, 101164
May 13, 2026 © 2026 The Authors. Published by Elsevier Inc.

CellPress

## Article

# Discovery and characterization of gene-by-environment and epistatic genetic effects in a vertebrate model

Bettina Welz,[1,2,8] Saul Pierotti,[3,8] Tomas Fitzgerald,[3] Thomas Thumberger,[1] Risa Suzuki,[1,2] Philip Watson,[1,2] Jana Fuss,[1,2] Tiago Cordeiro da Trindade,[1] Fanny Defranoux,[3] Marcio Ferreira,[3] Kiyoshi Naruse,[4] Felix Loosli,[5,*] Jakob Gierten,[1,6,7,*] Joachim Wittbrodt,[1,7,*] and Ewan Birney[3,9,*]

[1]Centre for Organismal Studies (COS), Heidelberg University, 69120 Heidelberg, Germany
[2]Heidelberg Biosciences International Graduate School (HBIGS), Heidelberg University, 69120 Heidelberg, Germany
[3]European Molecular Biology Laboratory, European Bioinformatics Institute (EMBL-EBI), Cambridge CB10 1SD, UK
[4]National Institute for Basic Biology, National Institutes of Natural Sciences, Okazaki 444-8585, Aichi, Japan
[5]Institute of Biological and Chemical Systems, Biological Information Processing (IBCS-BIP), Karlsruhe Institute of Technology, 76131 Karlsruhe, Germany
[6]Department of Pediatric Cardiology, Heidelberg University Hospital, 69120 Heidelberg, Germany
[7]German Centre for Cardiovascular Research (DZHK), Partner Site Heidelberg, Mannheim, Germany
[8]These authors contributed equally
[9]Lead contact
*Correspondence: felix.loosli@kit.edu (F.L.), jakob.gierten@cos.uni-heidelberg.de (J.G.), jochen.wittbrodt@cos.uni-heidelberg.de (J.W.), birney@ebi.ac.uk (E.B.)

## SUMMARY

Phenotypic variation arises from interactions between genetic and environmental factors, but disentangling these effects for complex traits remains challenging in observational cohorts like human biobanks. Model organisms with controlled genetic and environmental variation complement human studies in analyzing higher-order effects such as gene-by-environment (G×E) interactions, dominance, and epistasis. We utilized 76 medaka strains from the Medaka Inbred Kiyosu-Karlsruhe (MIKK) panel to compare heart rate plasticity across temperatures. An F2 segregation analysis identified 16 quantitative trait loci (QTLs), many exhibiting dominance, G×E, G×G, and G×G×E interactions. We experimentally validated four candidate genes, revealing temperature-sensitive heart rate effects. Finally, we simulated how genome-wide association study (GWAS) discovery power depends on statistical model choice. Our results suggest that the limited detection of non-additive effects in human GWASs stems from current study designs and sample sizes. This work demonstrates the value of controlled model organism studies for dissecting complex trait genetics and informing association study design.

## INTRODUCTION

Understanding the causes of phenotypic variation is a fundamental question in genetics, past[1] and present.[2] While statistical arguments[3] and practical observations[4–6] suggest that additive genetic effects can explain a large part of the heritability for many phenotypes in humans, ample evidence in model organisms,[7–15] plant breeding,[16–18] and animal breeding[19–25] suggests that non-additive effects such as gene-by-environment (G×E) and epistatic (G×G) interactions are critical for understanding the genetic architecture underlying complex traits. These effects are also important for other tasks in genetics, such as accurate phenotype prediction. In fact, even in human genetics, the importance of non-additive effects for realizing the goals of precision health is well recognized[26,27] and is at the heart of the field of pharmacogenomics.[28] Moreover, several examples of environmentally dependent effects

in humans have been reported,[29–35] as well as other forms of non-additivity, such as dominance[36] and epistatic effects.[37] For example, Palmer et al.[36] identified 183 phenotype-locus pairs in the UK Biobank that show genome-wide significant dominance effects, while Currant et al.[37] reported epistatic interactions among common variants in the VSX2 and PRPH2 genes that affect photoreceptor cell layer thickness in the human eye.

Genome-wide association studies (GWASs) in humans have been a resounding success,[38] fueled by the establishment of large-scale cohorts for genetic research.[39–45] However, human studies are limited in most cases to an observational approach, with major confounding deriving from population structure and genotype-environment correlations.[46–48] In contrast, studies on model organisms and plant or animal breeding genetics has a long history of leveraging controlled experimental settings to dissect the origins of phenotypic

**CellPress**

**Cell Genomics**
Article

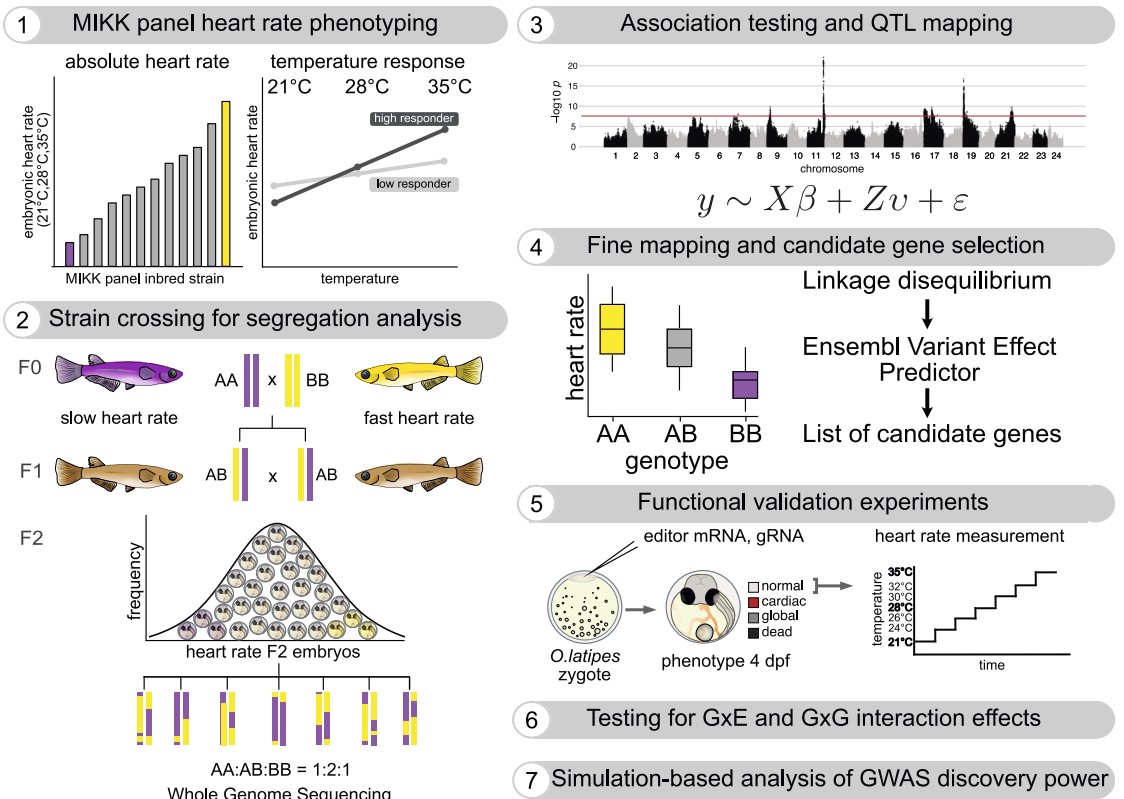

**Figure 1. Schematic of the study design for the identification and validation of genetic variants underlying embryonic heart rate differences at changing temperature conditions**

Steps 1, 2, 4, and 5 are illustrative and not based on experimental data. (1) The heart rate spectrum of genetically diverse MIKK panel inbred strains was assessed 4 days post fertilization (dpf) at 21°C, 28°C, and 35°C. (2) Eight strains differing in baseline heart rates and temperature responses were crossed to segregate phenotypic differences in an F2 population. (3) Association testing and quantitative trait locus (QTL) mapping identified genetic variants linked to heart rate divergence. (4) Fine mapping aided the identification of genes harboring variants with a predicted functional effect. (5) Gene editing was used to validate identified candidate genes and assess their temperature-dependent impact on heart rate. (6) Gene-by-environment (G×E) and epistatic (G×G) interaction effects were tested for their contribution to heart rate variation across the measured temperatures. (7) Simulations based on the experimental data reveal how model misspecification affects discovery power of association studies.

variation. Notable mentions are multi-parent advanced generation intercross (MAGIC) populations in plants,[49–52] laboratory and wild strains in yeast,[53–57] structured populations in worms,[58–61] the *Drosophila* Genetic Reference Panel (DGRP),[62,63] the mouse Collaborative Cross,[64–66] and heterogeneous stocks in rodents.[67,68]

Studies specifically designed to identify non-linear effects and to account for environmental influences on complex traits in model organisms indicate great complementary potential.[69–72] Mice in particular, as mammals, have a high translational relevance for human biology due to genetic, anatomical, and physiological similarities. However, the genetic variation that is covered by the available laboratory mouse strains[64,73–76] with complex domestication histories[77,78] does not reflect genetic variability in the wild. Thus, extending the analysis of complex traits to a vertebrate system that reflects the genetic diversity of a wild population promises to provide further insights into natural genetic variation.

In this study, we used medaka (*Oryzias latipes*), a well-established vertebrate model, to investigate G×E and epistatic

(G×G) interactions affecting the embryonic heart rate in the context of wild-derived haplotypes (Figure 1). Medaka has long been used as a genetic model system, with numerous wild-derived inbred strains and molecular tools available.[79–82] Its high fecundity, short generation time, small 700 Mb genome, and cost-effective husbandry make it particularly suitable for large-scale genetic studies. Moreover, its *ex utero* embryonic development allows for non-invasive extraction of physiological markers *in vivo* due to embryo transparency, facilitating phenotyping with scalable numbers.[83,84] Here, we leveraged the Medaka Inbred Kiyosu-Karlsruhe (MIKK) panel—a collection of 80 inbred strains derived from a wild population[85]—to study naturally occurring genetic and phenotypic variation in embryonic heart rate under controlled temperature conditions. Using a multi-parent F2 cross design, we identified 16 genetic loci driving heart rate differences and explored G×E and G×G interactions, some of which we could confirm by gene editing. Furthermore, we used simulations informed by our experimental results to explore the discoverability of genetic loci in humans.

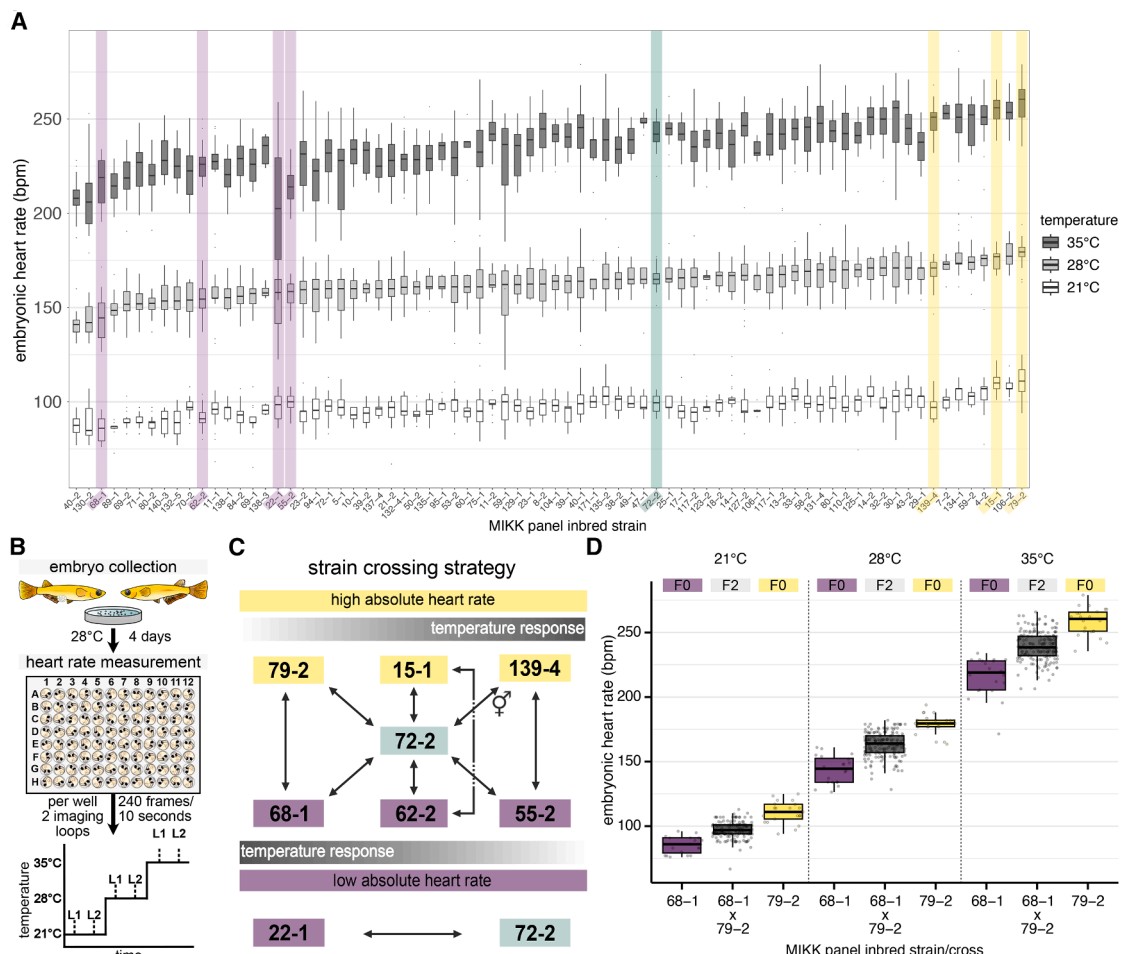

**Figure 2. Embryonic heart rate spectrum of MIKK panel strains at 21°C, 28°C, and 35°C and heart rate segregation through strategic crossing**

(A) Strain-dependent heart rate differences in beats per minute (bpm) for 76 MIKK inbred strains 4 dpf at 21°C, 28°C and 35°C. Strains were ranked by their median baseline heart rate at 28°C ($n$ = 5–47 embryos; Data S1). Strains selected for segregation analysis are highlighted (slow heart rate, varying temperature response, purple; average heart rate, green; fast heart rate, varying temperature response, yellow). Boxplots show data with median and 25th–75th percentile range.

(B) Experimental workflow for embryonic heart rate assessment 4 dpf. Images were acquired after a 20 min acclimatization period at the respective temperature in two consecutive imaging loops (L1 and L2) from which the average heart rate was calculated.

(C) Strategic crosses of eight MIKK panel strains with different heart rate phenotypes across temperatures provided the basis for segregation analysis and genetic mapping. The reciprocal crosses (72-2×139-4 and 139-4×72-2) are indicated using a combined male/female symbol.

(D) Heart rate of recombinant F2 embryos (gray) spread between parental (F0; purple and yellow) values across the measured temperatures, shown for one representative cross (for all crosses cf. Figure S1B). Boxplots show data with median and 25th–75th percentile range and overlaid scatterplots of individual heart rates in beats per minute (bpm). Sample sizes ($n$) for 21°C, 28°C, and 35°C: 68-1×79-2 F2 embryos $n$ = 180, 181, 179; 68-1 F0 $n$ = 20, 20, 21; 79-2 F0 $n$ = 21, 22, 22.

## RESULTS

### Temperature-dependent heart rate variation of MIKK panel inbred strains is under genetic control and can be dissected by a segregation analysis

We first assessed genetic contributions to heart rate variation under ecologically relevant temperature conditions by measuring embryonic heart rates of 76 wild-derived medaka inbred strains at moderate (21°C and 28°C) and more extreme (35°C) temperatures (Figures 2A and 2B); 4 MIKK strains were not fecund enough to support this assessment. Heart rates of 1,671 embryos were measured 4 days post fertilization (dpf) in

a highly randomized, and high-throughput setup, with 5–47 individuals phenotyped per strain. We observed that the median heart rate of the strains differed by as much as 26 beats per minute (bpm) (31% difference) at 21°C, 39 bpm (27% difference) at 28°C, and 58 bpm (29% difference) at 35°C. In addition to strain-dependent absolute heart rate differences, the magnitude of temperature-induced heart rate change also varied across the strains (Figure S1A). For all the three temperatures, we observed significant differences in heart rate between strains (Kruskal-Wallis $p < 2.2 \times 10^{-16}$).

To estimate the broad-sense heritability of the phenotype, we fitted ANOVA models stratified by temperature. The estimated

heritability was 0.42 (95% confidence interval [CI]: 0.40–0.49) at 21°C, 0.45 (95% CI: 0.43–0.51) at 28°C, and 0.44 (95% CI: 0.43–0.51) at 35°C. To identify the genetic variants contributing to these trait differences, we conducted a segregation analysis for genetic mapping. We selected eight isogenic MIKK panel strains with differing heart rate and temperature response characteristics (Figures 2A and S1A). In total, we performed 11 crosses (Figure 2C); one cross was carried out reciprocally by using maternal and paternal fish from the two founding strains, respectively. From each cross, we set up mating groups with the heterozygous F1 offspring to establish F2 populations. We assayed the heart rates at the same three temperatures (Figures 2D and S1B) and generated shallow whole-genome sequencing (WGS) data for 2,209 F2 individuals.

We previously described the WGS of this F2 population.[86] As expected, we observed that F2 individuals deriving from the same founder strains exhibited higher degrees of genetic relatedness compared to F2 individuals from independent crosses (Figure S2). Consistent with our F0 inbred strain assessment, mixed model-based heritability estimates for heart rate were determined to be 0.35 at 21°C (95% CI: 0.28–0.42), 0.46 at 28°C (95% CI: 0.40–0.53), and 0.45 at 35°C (95% CI: 0.38–0.52). Heritability for the difference in heart rate across temperatures (which we call the temperature response phenotype) was 0.31 between 28°C and 21°C (95% CI: 0.25–0.38), 0.34 between 35°C and 21°C (95% CI: 0.28–0.41), and 0.33 between 35°C and 28°C (95% CI: 0.27–0.40).

### Association testing reveals 16 QTLs linked to heart rate and pervasive interaction effects

A linear mixed model GWAS analysis of the combined genotype/phenotype F2 dataset revealed the presence of 16 quantitative trait loci (QTLs) that passed the phenotype-specific significance threshold (set by permutations) in at least one of the measured temperature conditions (Figures 3A and 3B; Table 1). Additionally, we ran the GWAS analysis on the difference in heart rate for each F2 embryo following temperature change (Figure S3). In these temperature response phenotypes, 7 of the 16 QTLs we observed in the stratified analysis remained detectable, while 9 were not. Given the large magnitude of the effect observed by a locus on chromosome 15 (chr15_qtl), we decided to include the lead SNP as a covariate before performing further analyses (Figures 3B and S3B). This also allowed for a clearer identification of the remaining loci. Overall, our complex F2 cross structure proved to be well powered to identify genetic associations with heart rate.

Our experimental design, which evaluated the heart rate of the same individuals across multiple temperature environments, allowed for direct investigation of G×E interactions. A qualitative example is the QTL detected on chromosome 15 (chr15_qtl). While this locus showed no association with heart rate at 21°C, it was significantly associated with heart rate at 28°C and 35°C. As expected, given this striking difference, chr15_qtl exhibited highly significant associations with all response phenotypes.

To quantitatively assess interaction effects and further characterize the genetic architecture of the heart rate phenotype, we evaluated both the significance and the proportion of variance

explained by different interaction terms in our dataset. For each QTL, heart rate measurements at 21°C, 28°C, and 35°C were analyzed jointly, and likelihood ratio tests were performed to compare linear models with or without the interaction term of interest. Relatedness and measurement temperature were accounted for using mixed models (STAR Methods). Multiple testing was controlled with a Bonferroni correction based on the total number of tests across QTLs and interaction modes (n = 48).

Overall, we observed that 8 of our 16 QTLs exhibit significant G×E effects, and 4 exhibit significant dominance effects (Figure 3C). In addition, 2 loci are significant for interactions between the dominance term and the temperature (which we call D×E). Of the 16 loci, only 7 are not significant for any interaction term (including dominance) and are thus fitting a purely additive model. In terms of variance explained, we notice a large difference among loci in the fraction of the overall phenotypic variance explained by the locus that is of purely additive origin (Figures 3D and 3E). chr15_qtl is the locus with the least linear effect, with less than 13% of the variance of additive origin. On the contrary, chr16_qtl is the locus that most closely resembles pure additivity, with almost 94% of the variance of additive origin. For the reciprocal cross, we found no loci showing parent-of-origin effects (STAR Methods; Data S9).

Besides looking at dominance and G×E interactions, we also assessed how different loci interact with each other in terms of epistatic (G×G) interactions (Figure 4A); the sample size of the multi-way F2 cross is powered to see major G×G effects between specific loci (STAR Methods for a power discussion). To control for the multiple testing burden, we restricted our analysis to interactions among the 16 QTLs identified in this study. We applied the same statistical framework used for the dominance and G×E tests described above (STAR Methods) and implemented a Bonferroni correction across QTLs and interaction modes (n = 240; 120 possible QTL pairs × 2 interaction modes). This correction was performed independent of the one applied to the dominance and G×E tests in Figure 3C.

We observed that 21 of 120 possible QTL pairs are significant for either G×G or G×G×E effects. Only 4 QTL pairs are significant uniquely for G×G effects, while the majority of the G×G interactions are themselves temperature dependent (G×G×E). 12 loci pairs are uniquely significant for G×G×E interactions but not G×G, while 5 locus pairs are significant for both G×G and G×G×E interactions. In this case, a locus being significant for G×G×E but not for G×G indicates that the marginal G×G effect is undetectable on its own but that the epistatic interaction becomes evident when the environmental context is considered.

### Experimental validation of identified candidate genes by gene editing phenocopy heart rate differences across temperatures

To experimentally test whether the identified variants are causal and influence heart rate in response to temperature, we selected five candidate genes: atg7, ccdc141, ppp3cca, ryr2b, and sptbn1 for functional validation experiments (Figure 5; Table 2). For the candidate gene selection, we focused on genetic variants that are in strong linkage with the lead SNP and have effect predictions in the coding regions with severe consequences (e.g., premature

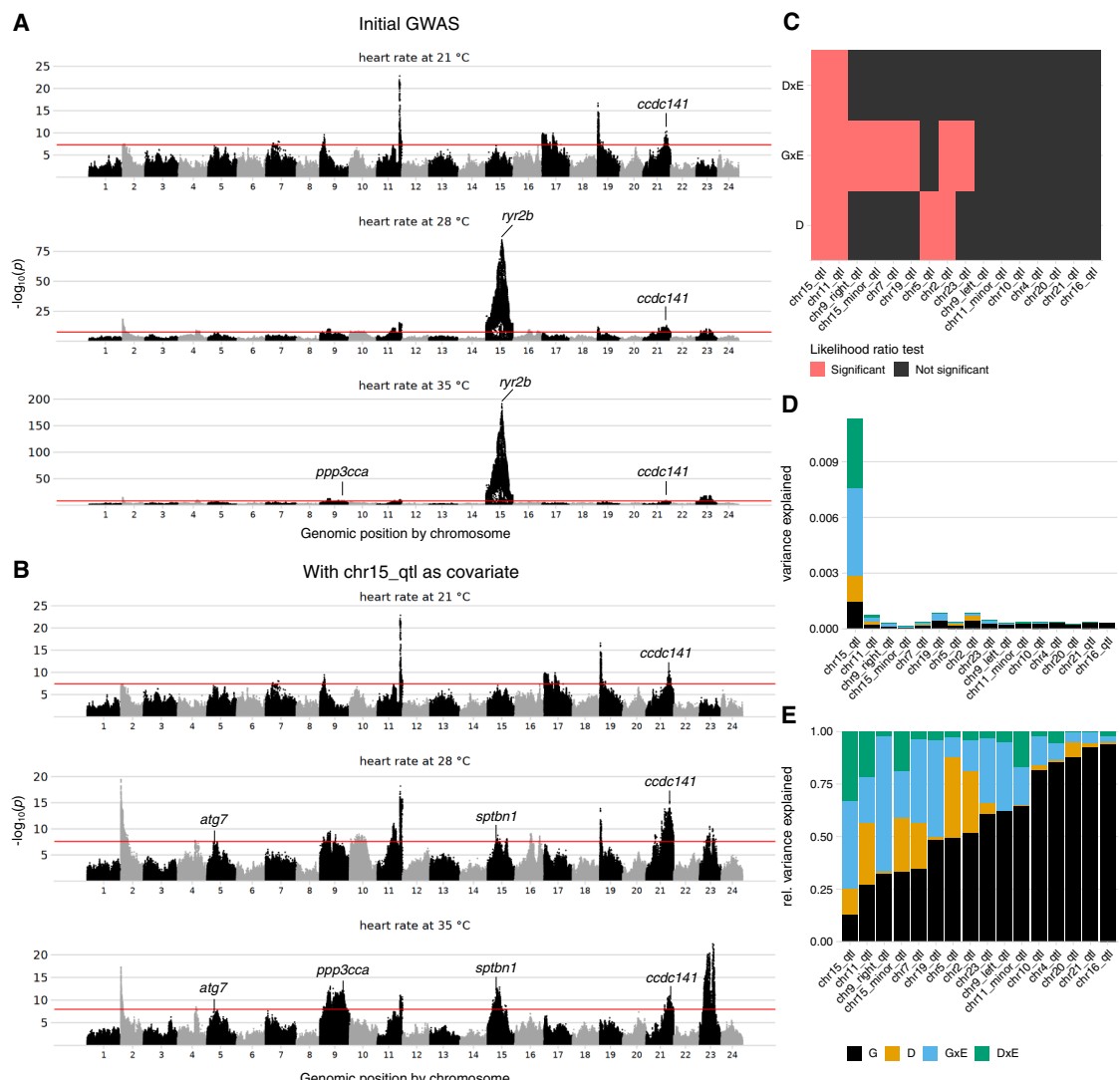

**Figure 3. GWAS results reveal 16 QTLs and G×G/G×E interaction effects underlying heart rate variance**

(A) Manhattan plot for the association of heart rate with genetics, stratified by environment (temperature). The significance threshold (red line) corresponds to the minimum *p* value achieved over 100 permutations. Genes selected for experimental validation are indicated.

(B) As in (A) but including the chromosome 15 locus as a covariate to reveal weaker associations.

(C) Heatmap of significant test results for dominance (D), G×E interactions, and dominance-by-environment interactions (D×E) terms per discovered loci.

(D) Percentage of variance explained by genetic (G; black), dominance (D; orange), G×E (blue), and dominance-by-environment (D×E; green) terms across the 16 discovered loci.

(E) Same as (D) but showing the relative variance explained as a proportion of the total variance attributable to the G, D, G×E, and D×E terms. *p* values: likelihood-ratio test.

stop codon or frameshift mutations; Data S2) and examined related phenotypic descriptions (STAR Methods). The *ccdc141*, *ryr2b*, and *sptbn1* genes are linked to cardiac traits (discussion). We performed gene perturbation using CRISPR-Cas9 or a cytosine base editor by microinjecting the editor mRNA and guide RNA (gRNA) into 1-cell stage embryos of the Cab medaka strain, a common wild-type reference. The choice of gene editing tool (base editor or CRISPR-Cas9) for functional validation experiments was made in a locus-specific manner (STAR Methods). Hereafter, we refer to the embryos edited using CRISPR-Cas9

as "crispants" and to the embryos edited using a base editor as "editants." In a two-step phenotyping process of the edited embryos, we examined effects on embryo morphology and heart rate 4 days post injection (dpi).

4 dpi, editing of the selected candidate genes resulted in an increased proportion of embryos with heart-specific morphological effects (9%–30%) and global phenotypes (16%–45%) compared to the mock-injected control group (heart-specific morphological effects, 5%; global phenotypes, 2%) (Figures S4A and S4C). The heart-affected embryos

**Table 1. Overview of the 16 mapped QTLs**

| Locus name | Genomic coordinates | Lead SNP position | Best model | *p* Value (best model) | *p* Value (model G) | Discovery phenotypes |
|---|---|---|---|---|---|---|
| *chr15_qtl* | 15:17,843,789−15:17,974,838 | 17,953,983 | G + E + D + G×E + D×E | 2.23E−308 | 1.68E−44 | 28°C; 35°C; diff. 35°C−21°C; diff. 28°C−21°C; diff. 35°C−28°C |
| *chr2_qtl* | 2:251,925−2:259,353 | 258,158 | G + E + D + G×E | 1.91E−24 | 4.43E−16 | 28°C; 35°C; diff. 35°C−21°C; diff. 28°C−21°C |
| *chr19_qtl* | 19:406,383−19:469,080 | 459,645 | G + E + G×E | 6.53E−24 | 5.82E−16 | 21°C; 28°C |
| *chr11_qtl* | 11:25,216,563−11:26,146,418 | 25,672,441 | G + E + D + G×E + DxE | 1.37E−20 | 6.16E−08 | 21°C; 28°C; 35°C |
| *chr21_qtl* | 21:26,755,000−21:26,814,684 | 26,634,956 | G + E | 1.01E−12 | 1.01E−12 | 21°C; 28°C; 35°C; diff. 28°C−21°C; diff. 35°C−21°C |
| *chr9_right_qtl* | 9:26,176,777−9:27,345,087 | 26,866,780 | G + E + G×E | 2.02E−12 | 1.85E−05 | 35°C; diff. 35°C−21°C; diff. 28°C−21°C; diff. 35°C−28°C |
| *chr23_qtl* | 23:15,202,259−23:15,220,660 | 15,220,347 | G + E + D + G×E | 1.87E−11 | 2.97E−09 | 28°C; 35°C; diff. 35°C−21°C; diff. 28°C−21°C |
| *chr10_qtl* | 10:7,320,585−10:10,587,092 | 8,415,034 | G + E + G×E | 1.09E−09 | 1.51E−09 | 21°C; 28°C |
| *chr5_qtl* | 5:11,355,552−5:11,654,482 | 11,500,853 | G + E + D | 1.27E−09 | 1.15E−06 | 28°C |
| *chr7_qtl* | 7:7,993,718−7:8,867,120 | 8,374,402 | G + E + D + G×E | 1.30E−09 | 2.25E−06 | 21°C |
| *chr4_qtl* | 4:19,263,287−4:20,825,589 | 19,720,609 | G + E | 1.50E−09 | 1.50E−09 | 28°C; 35°C |
| *chr9_left_qtl* | 9:4,860,591−9:5,515,585 | 5,045,398 | G + E | 5.03E−09 | 5.03E−09 | 21°C; 28°C |
| *chr16_qtl* | 16:27,252,665−16:28,804,411 | 28,190,196 | G + E | 7.75E−09 | 7.75E−09 | 21°C; 28°C |
| *chr15_minor_qtl* | 15:9,867,891−15:14,718,043 | 14,564,684 | G + E + G×E | 8.41E−09 | 6.31E−07 | 28°C; 35°C; diff. 35°C−21°C |
| *chr20_qtl* | 20:16,634,058−20:18,752,109 | 16,784,348 | G + E | 1.17E−07 | 1.17E−07 | 35°C |
| *chr11_minor_qtl* | 11:26,137,327−11:27,081,859 | 26,447,354 | G + E | 1.62E−07 | 1.62E−07 | 21°C; 28°C; 35°C; diff. 28°C−21°C |

The best model is determined by likelihood ratio tests against a covariate-only model (STAR Methods). Locus boundaries were manually determined by inspecting the association profiles. The lead SNP was usually the strongest associated, but a different SNP was chosen if it lacked a homozygous state. *p* values come from models fitted across all temperatures, not from the whole-genome discovery run (STAR Methods). The "Discovery phenotypes" column lists phenotypes where each QTL was significant in the whole-genome discovery run for each temperature (21°C, 28°C, and 35°C) and for heart rate changes between different (diff.) temperatures (35°C–21°C, 28°C–21°C, and 35°C–28°C). For all loci except for *chr15_qtl*, the lead SNP of *chr15_qtl* and its interactions with temperature were included as covariates. *p* values: likelihood-ratio test.

showed looping defects, pericardial edema, resized atrial or ventricular chambers, arrhythmia, and dextrocardia (Figure S4B). To determine whether the heart rate profiles of the edited embryos were altered in a temperature-dependent context, we exposed injected embryos to an expanded temperature ramp. We assessed heart rate changes in morphologically wild-type crispants and editants (and did not consider cardiac and globally affected embryos). Editing of the selected candidate genes revealed significant heart rate effects for four of the five tested loci (Figure 5B). While editing of three genes

(*ccdc141*, *ppp3cca*, and *sptbn1*) led to a heart rate increase, a heart rate decrease was observed for the *ryr2b* editants. For the *ryr2b* and the *ccdc141* genes, we identified a putative causal allele (a stop-gain and frameshift). In these two cases, the gene editing effect recapitulated the effect direction expected from the QTL discovery phase. We detected a temperature-independent genetic effect for *ryr2b* (*chr15_qtl*), *ccdc141* (*chr21_qtl*), and *sptbn1* (*chr15_minor_qtl*) and temperature-dependent genetic effects for *ryr2b*, *ccdc141*, *ppp3cca*, and *sptbn1* at a 5% false discovery rate. Some degree of

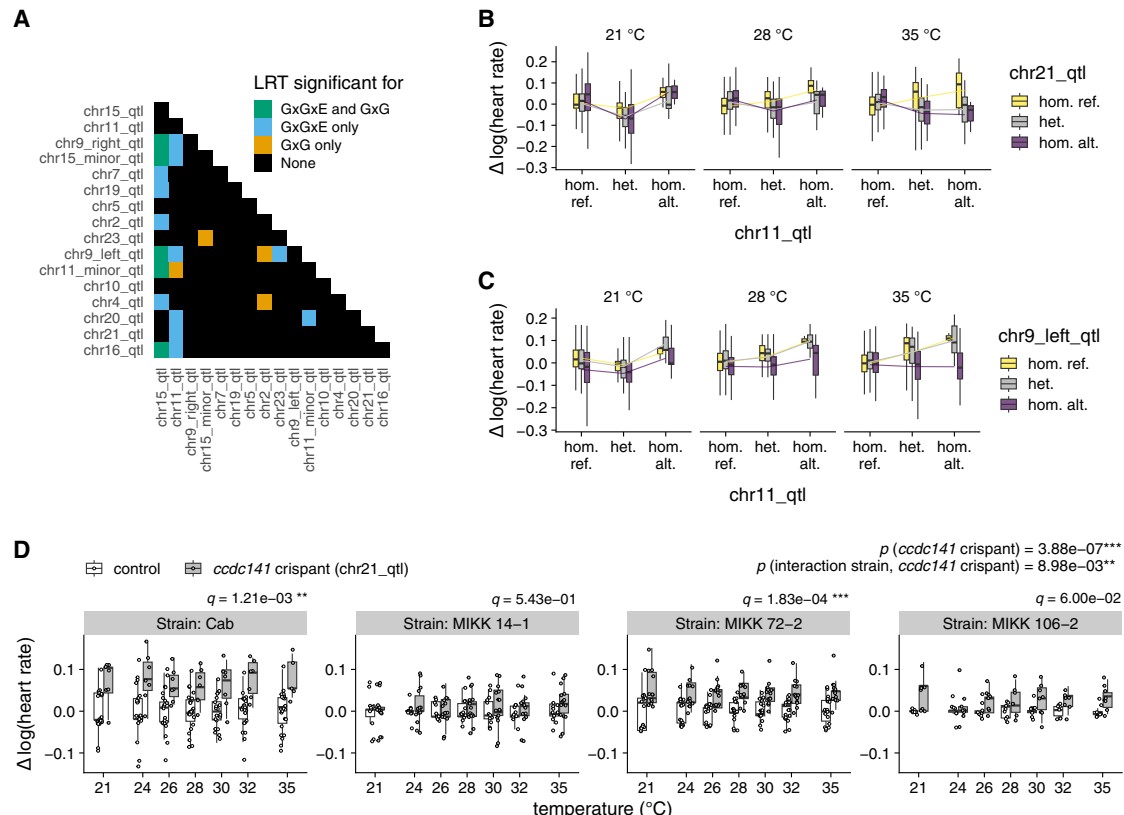

**Figure 4. Detection and experimental investigation of G×G and G×G×E interaction effects**

(A) Heatmap of pairwise G×G and G×G×E interactions among discovered loci. The lead SNP at *chr15_qtl* was included as a covariate to account for its large effect. *p* and *q* values: mixed-effect models.

(B) Interaction among *chr11_qtl* and *chr21_qtl*. The effect of *chr21_qtl* reverses depending on the genotype of *chr11_qtl* but only at 28°C and 35°C.

(C) Interaction among *chr11_qtl* and *chr9_left_qtl*. The effect of *chr9_left_qtl* is suppressed by homozygous reference *chr11_qtl* genotype, particularly at higher temperatures.

(D) Heart rates of *ccdc141* crispants (*chr21_qtl* candidate gene; *n* = 6–15 per strain; cf. Data S5 and S6) versus mock-injected controls (*n* = 7–23 per strain; cf. Data S5 and S6) vary with temperature and the genetic background of the injected individuals. *y* axis values are log transformed and normalized to the control mean per temperature. Boxplots show data with median and 25th–75th percentile range and overlaid scatterplots of individual heart rates. *p* values are reported for the general *ccdc141* editing effect on heart rate as well as within and between strains across the tested temperatures.

inter-individual variability in the effect size among the crispants and editants (injected generation) was observed for all candidate genes. This is likely due to genetic mosaicism in the injected generation, and it was statistically accounted for as a random effect for the individual embryo.

In summary, we experimentally confirmed functional effects on the heart rate in response to temperature for four of the five investigated candidate genes. The gene editing effects observed were broadly consistent in both direction and temperature dependence, with the corresponding QTL effects identified in the F2 cross GWAS.

To further characterize the candidate genes associated with heart rate in medaka, we analyzed their expression patterns by whole-mount *in situ* hybridization at embryonic stage 32, where heart rate was also measured. While some of the candidate genes were specifically expressed in the heart (*ccdc141* and *ryr2b*) indicating a local mechanism of action, others did not show cardiac expression (*sptbn1* and *ppp3cca*) (Figure 5C). The gene *atg7* showed a ubiquitous expression pattern, whereas

*sptbn1* was ubiquitously expressed except in heart tissue. Interestingly, the candidate gene *ppp3cca* was expressed in the habenula, a part of the brain, and in the retinal pigment epithelium in the eyes, both suggesting a secondary effect on heart rate. The different expression domains of the candidates identified by genetic mapping suggest diverse cellular and molecular mechanisms influencing cardiac function across the MIKK panel strains.

Since heart rate is a complex trait that can be influenced by multiple factors, we next sought to experimentally investigate the phenotypic consequences of an identified variant, taking into account possible genetic interactions. To experimentally investigate how epistatic (G×G) interactions might influence phenotypic penetrance, we targeted, in different genetic backgrounds the candidate gene *ccdc141*, for which an environmentally dependent G×G interaction (G×G×E) was detected in the F2 GWAS data (Figure 4A). To achieve this, we injected the CRISPR-Cas9 reagents into different medaka strains (Cab strain and MIKK panel strains 14-1,

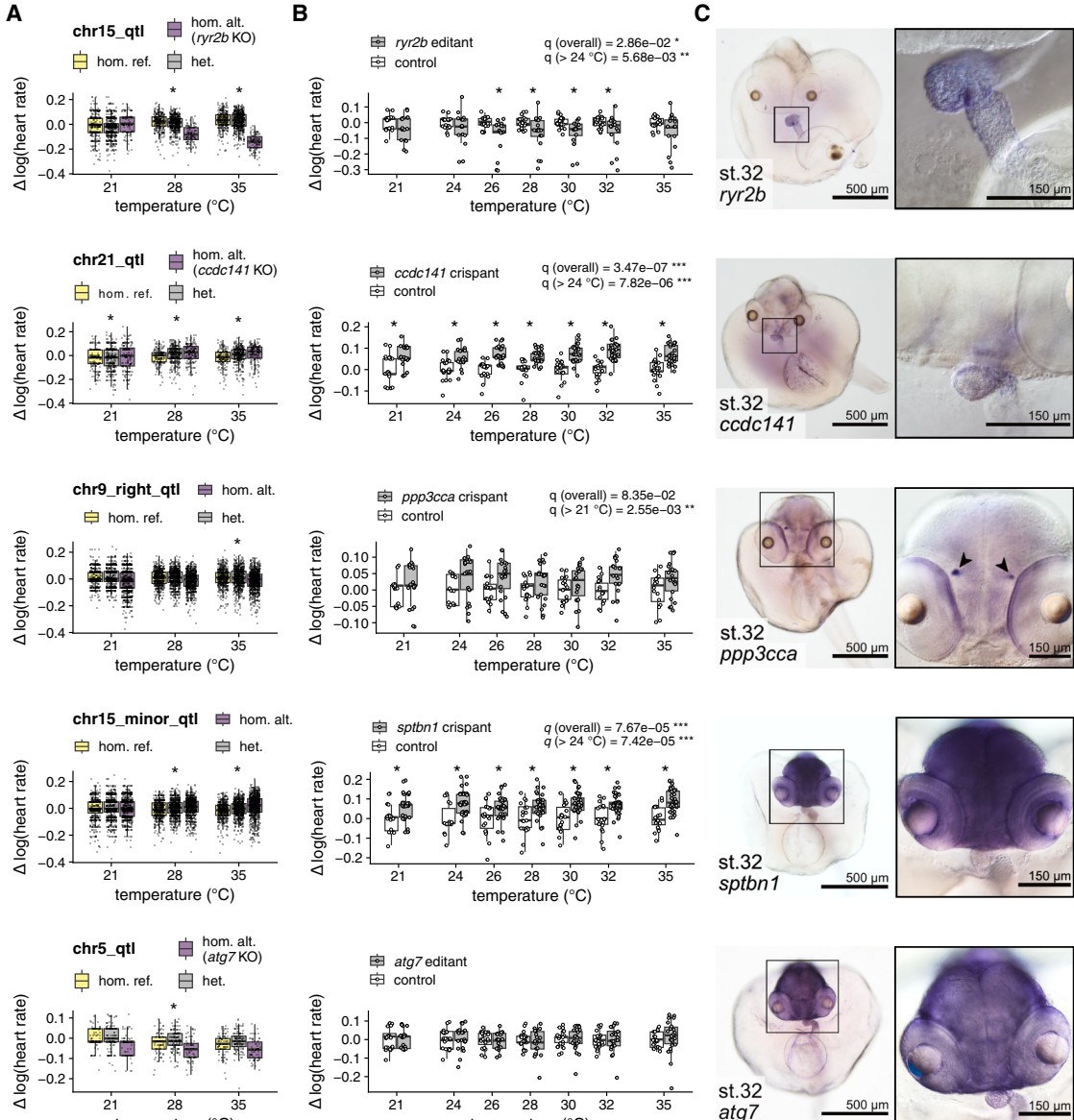

**Figure 5. Experimental characterization of candidate genes confirms temperature-dependent heart rate differences through gene editing and reveals their expression domains**

(A) Boxplot showing the genotype effect on F2 embryo heart rates across temperatures for 5 selected QTLs. Genotypes: homozygous reference (yellow), heterozygous (gray), homozygous alternate (purple). To aid visualization, datapoints were mean centered across temperatures and log transformed.

(B) Heart rate assessment 4 dpi of normally developed crispants (gray, $n$ = 21–38; cf. Data S5 and S6) and editants (gray, $n$ = 17–23; cf. Data S5 and S6) compared to mock-injected controls (white, $n$ = 15–24; cf. Data S5 and S6) across 21°C–35°C. $q$ values (mixed-effect models) are reported for the presence of a temperature-independent (G) and temperature-dependent (G×E) effect (STAR Methods). The $y$ axis is log transformed and centered on the mean of the control samples for each temperature. Boxplots show data with median and 25th–75th percentile range and overlaid scatterplots of individual heart rates. Linear models were also individually fit separately for each temperature. Individual comparisons that are significant at 5% false discovery rate (FDR) are denoted with a star (mixed-effect models).

(C) Whole-mount *in situ* hybridization showing expression of candidate genes at stage 32 in the albinism Heino strain.

72-2, and 106-2). While editing of *ccdc141* robustly increased the heart rate in the Cab strain and in the MIKK panel strain 72-2, we were unable to detect a significant heart rate change upon editing of *ccdc141* in the MIKK panel strains 14-1 and 106-2 (Figure 4C). The strain dependence of the genetic effect

is statistically significant ($p$ = 8.98*10$^{-3}$), providing evidence of a G×G interaction between the *ccdc141* edit and other, unidentified genetic variants that differ across strains. However, it should be noted that this experiment does not validate the presence of any specific epistatic pair. Rather, it demonstrates

**Table 2. Overview of the five candidate genes selected for experimental validation, QTL, variants, tissue of expression, and wild allele frequencies**

| Candidate gene | Locus name | Putative causal variant position | VEP consequence | Tissue of expression | Wild allele frequency causal variant | MIKK panel allele frequency |
|---|---|---|---|---|---|---|
| *ryr2b* | *chr15_qtl* | 18,044,484, 18,043,338, 18,006,656 | stop gained, frameshift, stop gained | heart | 0.030, not observed, 0.008 | 0.027, 0.007, 0.02 |
| *ccdc141* | *chr21_qtl* | 26,749,290 | frameshift | heart | 0.005 | 0.05 |
| *ppp3cca* | *chr9_right_qtl* | – | – | retinal pigment epithelium, habenula | – | – |
| *sptbn1* | *chr15_minor_qtl* | – | – | ubiquitous except heart | – | – |
| *atg7* | *chr5_qtl* | 9,430,242 | stop gained | ubiquitous | 0.202 | 0.14 |

that the effect of genetically altering *ccdc141* is modulated by the strain's genetic background.

Having functionally validated the selected candidate genes, we next investigated whether the genetic variants identified by association mapping are present in the wild. We sampled medaka fish from natural locations in close proximity to the original MIKK panel sampling site. For the two loci with putative causal alleles giving rise to disabled genes (*ryr2b* and *ccdc141*) confirmed with equivalent gene editing, we observed disabling variants in the wild at the same location as the F2 cross in both cases (Table 2). This confirmed that the mapped MIKK panel variants still occur naturally and are of ecological relevance.

## Simulations highlight the effect of model misspecification on GWAS discovery power

The pervasive presence of interaction effects in our dataset prompted us to investigate the consequences of ignoring these terms in a setting akin to a human GWAS. To this end, we simulated synthetic SNPs and environmental terms with additive, dominance, and interaction effect sizes estimated from our well-powered medaka genotype and phenotype data but in the context of an unrelated outbred population. This simulation is, by necessity, a simplification of real-world data. To demonstrate its applicability to outbred populations, including humans, we applied the same framework to human height (Figure S5). The results suggest that, at least for purely genetic effects, our simulations are appropriate provided the trait of interest has effect sizes and residual variance comparable to those observed in medaka.

We tested generative models both with and without epistatic (G×G and G×G×E) effects. Epistatic effects were evaluated with respect to two loci—*chr15_qtl* and *chr21_qtl*—for which we confirmed the frequency of the causal variant in the wild. Using this synthetic dataset, we determined the consequence of performing a GWAS using statistical models with varying degrees of misspecification. These discovery models ranged from the original data-generating model with additive G, D, G×E, D×E, G×G, and G×G×E interactions included to a simple additive-only SNP model with no knowledge of the environmental covariates. In addition, we investigated the effect of uncertainty in the environmental measurement, and we simulated the commonplace scenario of not directly genotyping the causal genetic variant but only a SNP tagging the causal variant.

For all of these conditions, we determined the minimum sample size required for the simulated locus to be discoverable at the $5 \times 10^{-8}$ significance threshold that is commonly used in human genetics.[87] We deem a simulated locus as discoverable under a given model if the required minimum sample size is smaller than 500,000 samples (approximative size of large human cohorts such as the UK Biobank) for a simulated minor-allele frequency smaller than 1%. The results of our simulation are presented in Figure 6 and summarized in Table S3.

First of all, we observe that directly genotyping the causal variant is critical for discovery. Of the 14 simulated loci, only 1 is discoverable with a G+E model when the genetic variant tested has a 0.9 $r^2$ correlation with the causal variant. On the contrary, when the discovery model has direct access to the causal variant, 12 of the 14 loci are discoverable. For higher allele frequencies, such as 10%, tagging approaches would discover the majority of loci. Measuring the environment also proved to be a large determinant of discovery power in our simulations; a model with no access to the environment cannot discover any of the tested loci. Even imperfect access to the environmental covariate, in the form of an environment measurement with 1% added noise, allows for the discovery of 11 of the 14 loci.

On the contrary, we notice that interaction effects can, in most cases, be safely omitted from discovery modeling without an appreciable increase in sample size requirements. Using the full model that generated the simulation, which includes G×E, D, D×E, G×G, and G×G×E terms, does not allow for the recovery of any additional locus compared to a G+E+G×E model. The inclusion of D or G×E effects plays an important role in reducing sample size requirements for only 2 of the 14 loci.

Additive effect size estimates depended on allele frequency. In the extreme scenario of overdominance, which we clearly observed in 2 of the 16 medaka QTLs, this can lead to reversal of the effect direction or complete non-discoverability of the locus. A clear example of this scenario is *chr5_qtl,* which, at an allele frequency of 1%, exhibits a sharp increase in required sample size. This is due to the dominance (overdominance in this case) and additive effects almost perfectly canceling out at this specific allele frequency, leading to an estimated effect size close to 0 for an additive-only model (Figure S7).

Non-additive effects, such as dominance and G×E interactions, were substantially more difficult to detect than additive effects in our simulations (Figure S8). Dominance was undetectable for the median locus at a 1% allele frequency, even with a

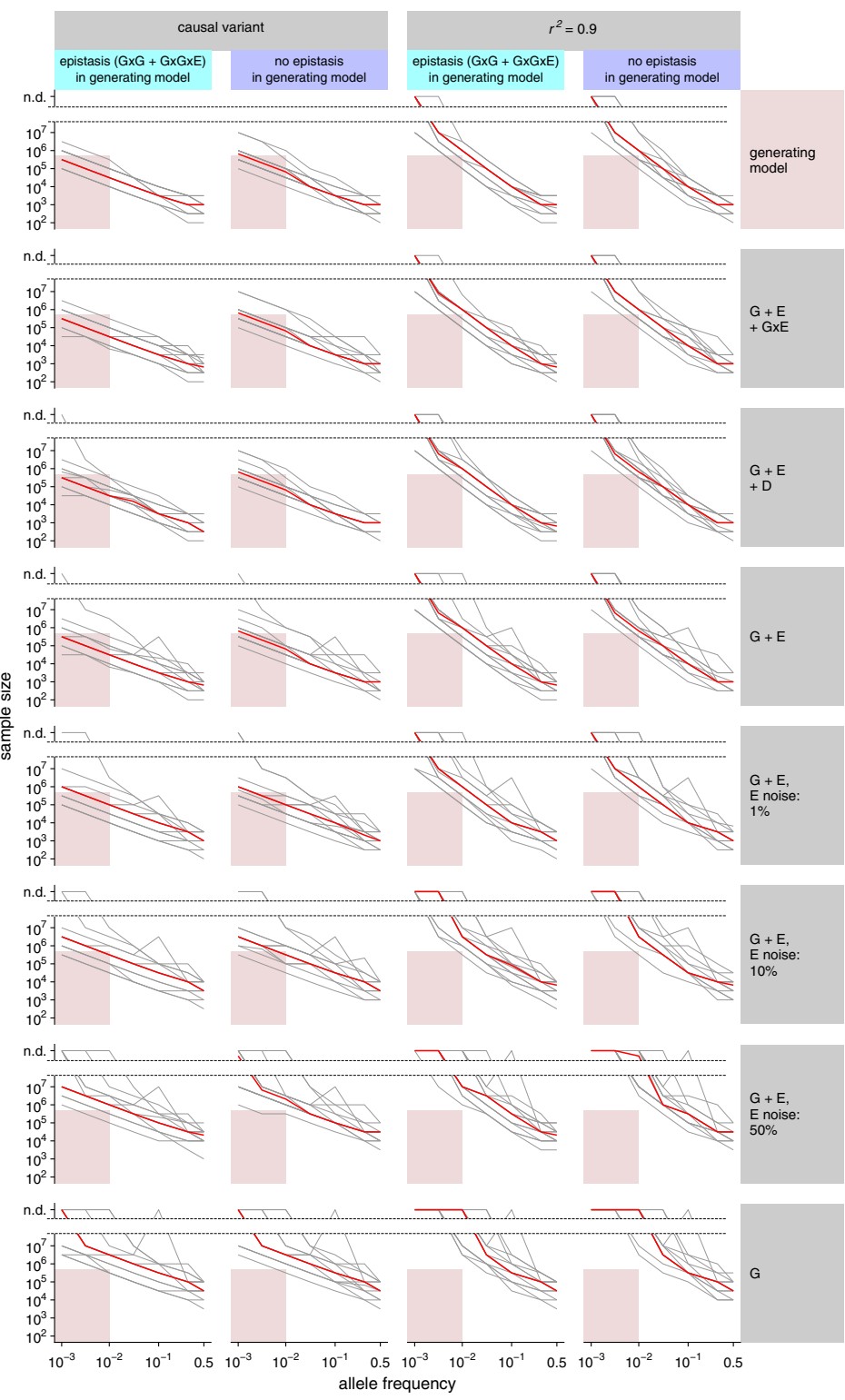

sample size of 10 million. G×E interactions, instead, were highly sensitive to measurement noise in the environmental variable; with 10% environmental noise, the median locus at 1% allele frequency remained undetectable even with 10 million samples. In contrast, with perfect environmental information, the G×E effect became detectable with fewer than 1 million samples. This implies that, even with current large human cohorts of 500,000–1 million individuals, we should not expect to detect dominance at most large-effect loci even if it is present. Similarly, interactions with the environment are only detectable when the causal variant is directly genotyped and the relevant environmental factor is accurately measured.

## DISCUSSION

Our study explored the genetic architecture of heart rate in the medaka fish vertebrate model in the context of experimentally controlled temperature variation (21°C–35°C). We leveraged 76 wild-derived inbred strains from the MIKK panel[85] to detect strain-dependent heart rate differences as well as differences in the heart rate response to temperature changes. The estimated trait heritability for the heart rate differences suggested an underlying genetic basis. By performing phenotype-genotype segregation experiments with eight founding strains, we discovered 16 QTLs affecting heart rate. For 4 of these QTLs, we were able to fine-map the causal gene and experimentally validate their cardiac relevance using gene editing tools and to characterize their spatial expression patterns. Of the 16 QTLs, 8 exhibit clear interaction effects with temperature (G×E), 4 exhibit dominance effects (D), and 2 display third-order interactions between dominance and temperature (D×E). We detected 21 epistatic (G×G) interactions among the QTLs, 17 of which exhibited significant third-order G×G×E effects with temperature. Using simulations based on our temperature-dependent heart rate QTL dataset in medaka, we investigated the genetic discovery power of an additive statistical model in the presence of widespread interaction effects and showed that it still works adequately for genetic discovery. However, our simulations demonstrated that genotyping the causal variant is a key determinant of discovery power in genetic association studies and that measuring the environment as precisely as possible is crucial for detecting genetic associations.

The 16 identified QTLs aided the experimental validation of four genes underlying heart rate differences in wild-derived medaka (*ryr2b*, *ppp3cca*, *ccdc141*, and *sptbn1*), which are highly conserved and have human orthologs or co-orthologs (Figure S9). *ryr2b* is co-orthologous to human RYR2, a classic cardiac calcium release channel known to play a fundamental role in heart function.[88] Genetic variants in ryanodine receptors (RyRs) are linked to many human diseases, and *RYR2* variants can cause catecholaminergic polymorphic ventricular tachycardia, leading to arrhythmia during physical or emotional

stress and even result in sudden cardiac arrest.[89–91] Interestingly, the channel conductance of sheep cardiac RyR has been shown to be temperature dependent.[92,93] Our data show that this also applies to the heart rate of *ryr2b* variant carrier strains from the MIKK panel, with the identified temperature-sensitive *ryr2b* variants also present in wild-caught medaka. *ccdc141* is orthologous to human CCDC141, also called CAMDI. This gene is involved in neuronal migration in mice[94] and is associated with heart rate variation in humans[95,96] and medaka.[97] *ppp3cca* is co-orthologous to the human calcineurin catalytic isozyme PPP3CC, which has been associated with blood pressure variation,[98,99] with calcineurin playing a key role in heart function.[100] Finally, *sptbn1* is co-orthologous to human SPTBN1, which has been associated with electrocardiogram features,[101,102] mitral valve prolapse,[103] hypoplastic left heart syndrome (HLHS),[104] and cardiovascular age.[105] In HLHS probands, SPTBN1 came into focus when the oligogenic disease ethology was investigated by testing for genetic interactions,[104] which we also detected for the QTL on which *sptbn1* is located in our dataset.

Collectively, the analyzed candidate genes validate use of the MIKK panel and our study approach for identifying human-relevant genetic variants associated with heart function and disease. Our experiments showed that the phenotypic penetrance and expressivity of these edited genes depended on both the temperature and the genetic background in which they were measured. Moreover, the embryonic heart rate in medaka has been shown to serve as an indicator of physical fitness and cardiac function in adulthood,[106] suggesting that the identified genetic variants may be relevant throughout life.

Our multi-parental F2 cross is well suited for discovering novel associations, achieving high allele frequencies even for large-effect alleles that are rare in outbred populations. For example, the *ryr2b* variant we discovered would have required a population size roughly six times larger to be detectable at its wild allele frequency of 0.03. As expected, however, fine mapping was challenging due to the limited number of recombination events. Consequently, causal variants were mainly identifiable in QTLs with severe loss-of-function mutations (*ryr2b* and *ccdc141*) and heart-related genes (*ryr2b*, *ccdc141*, and *sptbn1*).

Among our 16 identified QTLs, 50% exhibit significant G×E interactions, a proportion comparable to estimates in yeast for G×E effects on gene expression[57] and higher than estimates in flies.[70] In humans, a variance QTL study on cardiac phenotypes detected G×E effects in 22% of the loci tested.[107] As for epistatic interactions, we observed significant effects in 17.5% of locus pairs and a pervasive presence of G×G×E effects. This differs from reports in yeast, where G×G×E has been described as playing a less prominent role.[108] To avoid conflating statistical with biological interactions, we note that G×E effects may be scale dependent and reflect non-linear phenotype-mediator relationships rather than mechanisms.[109]

**Figure 6. Simulation of the factors affecting discovery power in outbred GWASs and of the consequence of model misspecification**
Shown is the minimum sample size required to discover ($p < 5 \times 10^{-8}$) a locus with effect sizes matching those found in medaka, across generating models (G×G yes/no, see main text), discovery models (genetics only, linear, interactions, and full model), environmental (E) measurement noise (0, 1%, 10%, and 50%), allele frequencies ($10^{-3}$ to 0.5), and genotype vs. causal variant correlation (causal variant, $r^2 = 0.9$). Allele frequency refers to the most recessive allele. Gray lines represent the median across 100 replicates per medaka QTL, and the red line represents the median of QTL medians ($n = 14$).

In our study, we mitigated such artifacts by applying a variance-stabilizing transformation to the phenotype and designing the experiment to prevent hidden genotype-environment correlations. We limited our analysis to pairwise epistasis among genome-wide significant loci due to the combinatorial complexity of testing many loci, especially for higher-order interactions involving three or more loci. Future studies of higher-order contributions may yield mechanistic insights but will face major challenges from multiple testing and the rarity of higher-order allele combinations. An alternative is to examine the same variant across medaka strains (Figure 4D), providing indirect evidence of epistasis with differing background variants.

As stated before, our simulations are a simplification of real-world data. Although we capture certain aspects of linkage disequilibrium by tagging SNPs, we do not model interactions between QTLs in linkage disequilibrium (e.g., neighboring loci) or interactions between population stratification and environmental factors, both of which are common sources of confounding in outbred populations. In addition, traits in medaka fish and humans may have distinct genetic architectures and combinations of confounders, further complicated by the temporal variability of many environmental measures. Accordingly, our simulations should be regarded as best-case scenarios for the discovery of individual QTLs rather than as comprehensive models of complex trait architecture in humans. Nevertheless, they remain informative about the parameter space relevant to outbreeding studies.

Notably, when the causal variant is directly tested, an additive discovery model is generally suitable for discovering rare alleles (below 1% allele frequency), even in the presence of widespread interaction effects. However, using a proxy locus ("SNP tagging") or measuring the environment with >10% noise significantly impacts the discovery of loci at 1% allele frequency, while loci at 10% allele frequency remain detectable. Thus, the simulations identified two key determinants of statistical power: (1) reasonably accurate (<10% additional variance) measurement of environmental variables and (2) direct genotyping of causal variants through WGS rather than tagging variants. Consequently, we recommend prioritizing these aspects when designing GWAS cohorts, particularly in human studies, where environmental control is more limited than in model organisms.

As expected from theoretical considerations and previous work,[36] non-additive effects were notably difficult to detect in our simulations, particularly dominance, which often required sample sizes in the multiple millions at a 1% allele frequency. These findings align with previous work underscoring the limited contribution of non-additive effects to complex trait variation at the population level[3] as well as recent studies examining the role of dominance in human cohorts.[36] In our view, this study and the accompanying simulations indicate that there is no fundamental difference between human and non-human vertebrate trait architectures. While human studies have primarily identified and modeled additive effects, research in other species has leveraged controlled environments and breeding to uncover a broader range of trait-associated interactions. Although non-additive effects can be ignored for discovery or some population modeling, this does not mean they do not have important roles in individual predictions, in particular at

phenotypic extremes, as is already the case for the need to consider dominance/recessive effects in many rare disease diagnoses. Furthermore, as the environment can often be a modifiable factor, discovery of G×E interactions can provide not just improved prediction but also open up other intervention pathways. The same applies to epistatic effects, which we found to make only a minor contribution to QTL discoverability. Although such effects may be of large magnitude and critical for individuals carrying specific allelic combinations, their overall contribution to phenotypic variance—and thus to discoverability—likely remains small due to their combinatorial rarity.

Our simulation suggests that increasing sample size alone offers limited power for discovering interactions; alternative strategies, such as sampling bottleneck populations at reasonably large scale, where drift elevates allele frequencies, may be more effective.

One of the main advantages of working with model organisms such as medaka is the ability to control both environmental and genetic variation, which is crucial in dissecting the dynamics of G×E effects. For example, in this study, we leveraged the possibility to finely vary the temperature and genetically engineer medaka to demonstrate how the effect of a genetic variant affecting the *ryr2b* gene materializes only above a temperature threshold of 24°C. More broadly, medaka sits at an optimal intermediate level of complexity; as a vertebrate, it shares key physiological and genetic similarities with humans, but it is more cost effective on a large scale than mammalian models. Moreover, its high tolerance to inbreeding allows for the direct establishment of inbred strains from the wild. Studies in medaka are thus complementary to those in laboratory mice, which are more closely related to human and have been extensively studied. The MIKK panel we previously developed and this work exemplify how these advantages can be harnessed in practice. In the future, our approach to dissecting G×E and G×G interactions can be extended to more phenotypes relevant to human physiology; other environmental effects, such as exposure to chemical toxins or drugs; and other forms of genetic variation besides small nucleotide variation.

In summary, our results lay the groundwork for future research of the genetic determinants of complex phenotypes, reinforce the value of medaka as a powerful model system, and provide guidance for the design of cohort studies that capture the full spectrum of variation—from environmental factors to genetic diversity.

### Limitations of the study

Our study has some limitations. First, the GWAS mapping resolution was limited by the linkage disequilibrium structure. In future work, this could be improved by extending the breeding scheme beyond the second generation, performing backcrosses, increasing sample sizes, or additionally sampling outbred wild populations. Second, we investigated epistatic interactions by assessing the same genetic variant across different strain backgrounds, but we did not experimentally validate specific locus pairs. One potential approach to identify such interactions would be to cross inbred strains fixed for different alleles at multiple loci and test whether their combined phenotypic effects deviate from additivity. Third, the simulation (Figure 6) assumes similar effect

sizes and distributions of genetic and environmental variables between medaka and humans. If these assumptions are unmet, then human cohort power estimates should be interpreted cautiously. Our simulations are intended to provide guidance on the relative influence of study design on GWAS discovery power, not absolute estimates. Empirical validation using human datasets will be required to confirm the applicability to human cohorts.

## RESOURCE AVAILABILITY

### Lead contact

Requests for further information and resources should be directed to and will be fulfilled by the lead contact, Ewan Birney (birney@ebi.ac.uk).

### Materials availability

All unique/stable reagents generated in this study are available from the lead contact with a completed materials transfer agreement.

### Data and code availability

- Raw sequencing reads were deposited at the European Nucleotide Archive (ENA; https://www.ebi.ac.uk/ena/browser/home) under accession PRJEB90425.
- Medaka heart rate measurements are part of the supplementary datasets.
- Detailed statistical analyses results are part of the supplementary datasets.
- All original code has been deposited at the following repositories:
  - https://github.com/birneylab/varexplore
  - https://github.com/birneylab/heart_rate_tmp_gxe
  - https://github.com/birneylab/heart_rate_tmp_pairwise_gxg
  - https://github.com/birneylab/heart_rate_tmp_simulation_gxg
- Any additional information required to reanalyze the data reported in this paper is available from the lead contact upon request.

## ACKNOWLEDGMENTS

We thank all members of the Birney and Wittbrodt research group for valuable input to the project and the manuscript. We thank Tinatini Tavhelidse-Suck for contributing to the COS-MIKK propagation campaign; Rachel Müller, Tanja Kellner, and Beate Wittbrodt for expert technical assistance; and Rie Ajioka, Yukari Koike, Yuko Teshima, Nadeshda Wolf, Natalja Kusminski, Thomas Seitz, Marzena Liv, Erik Leist, and Antonino Sarazeno for animal husbandry. We thank Kouko Yamazaki for sampling wild medaka. We thank Jochen Gehrig and Laurent Thomas (ACQUIFER, Bruker) for technical advice on high-throughput imaging. We acknowledge Vladimir Benes, Ferris Jung, Mireia Osuna-López, and all members of the EMBL Genomics Core Facility (GeneCore) as well as the Wellcome Trust Sanger Institute for support with the library preparation and whole-genome sequencing. We thank the National Bioresource Project (NBRP) medaka Japan and NIBB individual collaborative research projects (24NIBB313) for access to the wild-caught medaka fish. This research was supported by the European Research Council Synergy Grant IndiGene (810172) and by the NIH (National Institutes of Health, R01ES029917).

## AUTHOR CONTRIBUTIONS

Conceptualization, B.W., S.P., K.N., J.G., F.L., J.W., and E.B.; data curation, B.W. and S.P.; formal analysis, B.W. and S.P.; funding acquisition, J.W. and E.B.; investigation, B.W., S.P., and T.C.d.T.; resources: B.W., T.T., R.S., P.W., J.F., and F.L.; software, S.P., T.F., F.D., and M.F.; visualization: B.W. and S.P.; writing – original draft, B.W., S.P., J.W., and E.B.

## DECLARATION OF INTERESTS

E.B. is a member of the *Cell Genomics* advisory board.

## DECLARATION OF GENERATIVE AI AND AI-ASSISTED TECHNOLOGIES IN THE WRITING PROCESS

During the preparation of this work, the authors used generative large language models (LLMs) to improve clarity and readability. After using this tool/service, the authors reviewed and edited the content as needed and take full responsibility for the content of the publication.

## STAR★METHODS

Detailed methods are provided in the online version of this paper and include the following:

- KEY RESOURCES TABLE
- EXPERIMENTAL MODEL AND STUDY PARTICIPANT DETAILS
  - Fish maintenance
  - Wild catches
- METHOD DETAILS
  - Automated image acquisition and heart rate detection
  - Crossing of phenotypic contrasting MIKK panel strains for segregation analysis
  - Genomic DNA extraction and sequencing library preparation of F1 and F2 individuals
  - Whole-genome sequencing and genotype imputation
  - gRNA design
  - *In vitro* transcription of mRNA
  - Microinjections
  - Genotyping of crispants and editants
  - Microscopy
  - Wholemount *in situ* hybridization
  - Locus definition and fine mapping
  - Candidate gene selection
  - The birneylab/varexplore pipeline
- QUANTIFICATION AND STATISTICAL ANALYSIS
  - Linear mixed model genetic association analysis
  - Cross-specific QTL detection
  - Detection of G×E and G×G effects
  - Calculation of variance explained by single loci
  - Statistical analysis of gene editing results
  - Simulation of the effects of model misspecification
  - Construction of gene trees for confirmed genes
  - Statistical tests for reciprocal cross effect
  - Human height UK Biobank simulation example
  - In addition, we provide the following list of assumptions and caveats

## SUPPLEMENTAL INFORMATION

Supplemental information can be found online at https://doi.org/10.1016/j.xgen.2025.101126.

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

## Article

**CellPress**

# STAR★METHODS

## KEY RESOURCES TABLE

| REAGENT or RESOURCE | SOURCE | IDENTIFIER |
|---|---|---|
| **Bacterial and virus strains** | | |
| Mach1T1™ T1[R] phage resistant chemically competent *Escherichia coli* | Thermo Fisher Scientific | Catalog #C862003 |
| **Chemicals, peptides, and recombinant proteins** | | |
| Q5 High-Fidelity DNA Polymerase | New England Biolabs | Catalog #M0491 |
| **Critical commercial assays** | | |
| mMessage mMachine Sp6 Transcription Kit | Thermo Fisher Scientific | Catalog #AM1340 |
| mMessage mMachine T7 Transcription Kit | Thermo Fisher Scientific | Catalog #AM1344 |
| Monarch DNA Gel Extraction Kit | New England Biolabs | Catalog #T1020 |
| RNeasy Mini Kit | Qiagen | Catalog #74,106 |
| **Deposited data** | | |
| Raw sequencing data for MIKK panel F2 crosses | This paper | ENA ID: PRJEB90425 |
| Medaka heart rate measurements | This paper | Data S3, S4 and S6 |
| Statistical test results | This paper | Data S7, S8 and S9 |
| **Experimental models: Organisms/strains** | | |
| Cab strain, strain background (*Oryzias latipes*) | Loosli et al.[110] | N/A |
| Heino strain, strain background (*Oryzias latipes*) | Loosli et al.[110] | N/A |
| MIKK strains, strain background (*Oryzias latipes*) | Fitzgerald et al.[85] | N/A |
| Wild catches from Kiyosu and in Maeshiba, strain background (*Oryzias latipes*) | This paper | N/A |
| **Oligonucleotides** | | |
| PCR primers, cloning primers, sequencing primers | Eurofins Genomics | N/A |
| Locus-specific crRNAs and the tracrRNA backbone | Integrated DNA Technologies | N/A |
| **Recombinant DNA** | | |
| Probe plasmid templates | Biocat; Souren et al.[111] | N/A |
| pCS2+ (evoBE4max) | Cornean et al.[112] | N/A |
| pCS2+ (heiCas9) | Thumberger et al.[113] | N/A |
| **Software and algorithms** | | |
| ACEofBASEs | Cornean et al.[112] | https://aceofbases.cos.uni-heidelberg.de |
| CCTop | Stemmer et al.[114] | https://cctop.cos.uni-heidelberg.de:8043/ |
| FEHAT | Ferreira et al.[115] | https://github.com/birneylab/FEHAT |
| Geneious Prime | Biomatters | Version 2019.2.3 |
| ImageJ | Schindelin et al.[116] | Version 2.3.0 |
| birneylab/flexlmm | Pierotti et al.[117] | https://github.com/birneylab/flexlmm |
| birneylab/varexplore | This paper | https://github.com/birneylab/varexplore https://doi.org/10.5281/zenodo.17609143 |
| GxE testing pipeline | This paper | https://github.com/birneylab/heart_rate_tmp_gxe https://doi.org/10.5281/zenodo.17609022 |
| GxG testing pipeline | This paper | https://github.com/birneylab/heart_rate_tmp_pairwise_gxg https://doi.org/10.5281/zenodo.17609003 |
| Simulation pipeline | This paper | https://github.com/birneylab/heart_rate_tmp_simulation_gxg https://doi.org/10.5281/zenodo.17608980 |

**Cell Genomics**
Article

## EXPERIMENTAL MODEL AND STUDY PARTICIPANT DETAILS

### Fish maintenance

All medaka (*Oryzias latipes*) strains were maintained at Heidelberg University and Karlsruhe Institute of Technology (fish husbandry, permit numbers 35–9185.64/BH Wittbrodt, AZ35–9185.64/BH KIT) in accordance with local animal welfare standards (Tierschutzgesetz §11, Abs. 1, Nr. 1) and with European Union animal welfare guidelines.[118] Fish were maintained as closed stocks in constant recirculating systems at water temperatures between 24°C and 26°C with a 14 h light/10 h dark cycle. The following medaka strains were used in this study: MIKK panel strains,[85] the Cab strain and the Heino strain.[110] Both sexes were used for experiments. The fish facility is under the supervision of the local representative of the animal welfare agency.

### Wild catches

To estimate the allele frequency of the mapped genetic variants in wild medaka populations, fish were sampled 2023 in Kiyosu and in Maeshiba near Toyohashi as previously described.[85,119] For whole genome sequencing, fin clips from 183 wild-derived medaka were collected at NIBB, Okazaki and sent to Heidelberg for extraction of genomic DNA with phenol-chloroform as described earlier.[86] Following DNA extraction, samples were sequenced at 16.8x to 33.2x coverage at the Wellcome Trust Sanger Institute as described earlier.[85]

## METHOD DETAILS

### Automated image acquisition and heart rate detection

F0 and F2 embryos were reared in hatching medium (2 mg/L Methylene Blue Trihydrate (Sigma-Aldrich) in 1x Embryo Rearing Medium (ERM) (17 mM sodium chloride, 0.4 mM potassium chloride, 0.27 mM calcium chloride dihydrate and 0.66 mM magnesium sulfate heptahydrate at pH 7) at 28°C for four days and medium was changed every other day. To ensure an unbiased analysis and maintain the full range of phenotypic diversity no filtering or selection of embryos besides their developmental stage was performed. For imaging, medaka embryos (stage 32, according to Iwamatsu[120]) were rolled on sandpaper to remove the outer surface hairs of the chorion, then transferred in 150 μL 1x ERM to a 96-U bottom well plate and then sealed using a semi-permeable foil. To acquire the heart rate information at controlled temperature conditions, automated imaging was performed in a 96-well plate format using the ACQUIFER imaging machine. Embryos were exposed to temperatures ranging from 21°C to 35°C. After an acclimation time of at least 20 min for each tested temperature, 240 images were acquired with 24 fps. The heart rate was measured within two consecutive imaging loops per temperature condition, and the heart rate average of the loops was used for further analysis. For all datasets, the heart rate information was extracted using the FEHAT detection tool.[115] FEHAT measurements below 50 bpm were not considered because they likely result from masked heart regions during image acquisition.

Automated heart rate acquisition for the crispant and editant embryos was carried out 4 days post injection in an extended temperature ramp with seven consecutive temperature treatments (21°C, 24°C, 26°C, 28°C, 30°C, 32°C, 35°C) using the ACQUIFER imaging machine. The following amendments have been made to the previously described imaging protocol: injected embryos were not rolled on sandpaper and to prevent large temporal shifts in extended temperature experiments only one imaging loop was acquired and analyzed per temperature condition.

### Crossing of phenotypic contrasting MIKK panel strains for segregation analysis

Eight MIKK panel strains with different temperature-dependent heart rate properties were selected to set up 11 different crosses (Table S1). The F1 generation was established from cross-mating one male with up to 5 female fish and the F2 embryos were collected from F1 families containing four to 40 individuals. In total 2667 F2 embryos were phenotyped and whole genome sequencing data were obtained for 2209 F2 embryos.

### Genomic DNA extraction and sequencing library preparation of F1 and F2 individuals

Genomic DNA extraction of F1 and F2 individuals and the sequencing library preparation was performed as previously described.[86]

### Whole-genome sequencing and genotype imputation

The F2 medaka embryos were sequenced and their genotype was imputed with the *birneylab/stitchimpute* pipeline as previously described.[86] The population analyzed in Pierotti et al.[86] is the same as the one used here, so readers can refer to that work for further details on the imputation process and validation. Briefly, 150 bp paired-end Illumina short-read sequencing was performed on a NextSeq2000 machine (https://www.illumina.com/). We sequenced a variable number of samples per flow cell, achieving an average sequencing depth of 1.4x. Reads were aligned to the Hdr-R medaka reference genome (ENSEMBL ID: ASM223467v1) using the *nf-core/sarek* pipeline.[121] As an aligner we used *bwa-mem2*[122] and reads were deduplicated with *GATK MarkDuplicates*.[123]

One F1 sample per cross and 2 F2 samples from the 72-2 x 55-2 cross were sequenced at higher depth (12 samples in total, obtaining a sequencing depth of 33x to 61x). These samples were used as a ground truth for validating the imputation process. These samples were also processed with the *nf-core/sarek* pipeline. As an aligner we used *bwa-mem2* and reads were deduplicated with

*GATK MarkDuplicates*. Genotypes were obtained with *GATK* via joint germline variant calling.[124] Raw sequencing reads were deposited at the European Nucleotide Archive (ENA, https://www.ebi.ac.uk/ena/browser/home) under accession PRJEB90425.

We ran the genotype imputation on the low-coverage F2 samples and high coverage F1 and F2 samples jointly. The high coverage samples were downsampled to a sequencing depth of 0.5x before being used for the imputation. Imputation accuracy was evaluated as the squared Pearson correlation ($r^2$) between the imputed genotypes of the downsampled high-coverage samples and the genotype calls obtained from the same samples with GATK.

The initial set of genetic variants used for the imputation process was obtained from the variants called by GATK on the high coverage samples and consisted of 6.2 million SNPs. We filtered this variant set with *bcftools view*[125] using the following filtering criteria: *-i 'CHROM != "MT"', -i 'TYPE == "snp"', -i 'N_ALT >= 1', -i 'MAC >= 1'*. This variant set was then refined iteratively using the *snp_set_refinement* mode of the *birneylab/stitchimpute* pipeline. We performed 5 iterations and at each iteration retained only the SNPs that passed a certain $r^2$ threshold. We used the following filter values: 0.5, 0.5, 0.75, 0.9, and 0.9. Our final set of SNPs consisted of 3.2 million variants.

Imputation parameters *K* and *nGen* were optimized using the *grid_search* mode of the *birneylab/stitchimpute* pipeline. We selected *K = 16* and *nGen = 2*. In addition, we used the following parameters: *expRate = 2, niterations = 100, shuffleHaplotypeIterations = seq(4, 88, 4), refillIterations = c(6, 10, 14, 18), shuffle_bin_radius = 1000*.

Overall, the imputed genotypes proved to be very reliable, with and average sample-wise $r^2$ value of 0.996 between the genotypes called on the high coverage samples using GATK and the imputation obtained from the same samples with reads downsampled to 0.5x depth.

### gRNA design

The gRNA target site was selected in close proximity to the mapped genetic variant and the gRNAs were designed as previously described with the target prediction tools *CCTop*[126] and *ACEofBASEs*.[112] Locus-specific crRNAs and the tracrRNA backbone were ordered via the Integrated DNA Technologies (IDT) synthesis service. The crRNA and the tracrRNA were both diluted in nuclease-free duplex buffer (IDT) to a final concentration of 40 μM and incubated at 95°C for 5 min to form a functional gRNA duplex (Table S2).

### *In vitro* transcription of mRNA

To generate the Cas9 mRNA and evoBE4max mRNA, pCS2+ (heiCas9)[113] and pCS2+ (evoBE4max)[112] were linearized by a NotI digest and the mRNA was transcribed using the mMESSAGE mMACHINE SP6 or T7 Transcription Kit (Thermo Fisher Scientific). RNA purification was performed with the RNeasy Mini Kit (Qiagen) according to manufacturer's instructions.

### Microinjections

For gene-editing, microinjections were performed at the 1-cell stage. The injection solution contained 150 ng/μL editor mRNA (either Cas9 or evoBE4max), 4 pmol gRNA and 10 ng/μL GFP mRNA as injection tracer. As a mock-control 10 ng/μL GFP mRNA was injected. Injected embryos were kept at 28°C in medaka embryo rearing medium (ERM, 17 mM NaCl, 40 mM KCl, 0.27 mM CaCl2, 0.66 mM MgSO4, 17 mM HEPES) and selected for GFP signal 7 h post injection. Phenotyping was done 4 days post fertilization.

For genes harboring a genetic variant with a stop gained predicted effect, we used the cytosine base editor evoBE4max to introduce a premature stop codon in close proximity to the originally observed variant (*ryr2b* and *atg7*). The CRISPR/Cas9 system was applied to investigate variants that resulted in a frameshift mutation (*ccdc141*), or when narrowing down to a single putative causal variant was not straightforward (*ppp3cca*, *sptbn1*).

### Genotyping of crispants and editants

Single embryos were lysed in DNA extraction buffer (0.4 M Tris/HCl pH 8.0, 0.15 M NaCl, 0.1% SDS, 5 mM EDTA pH 8.0; 1 mg/mL proteinase K) at 60°C overnight. Nuclease-free water was added 1:2 and the proteinase K was inactivated for 20 min at 95°C. To precipitate the genomic DNA, 300 mM sodium acetate and 3x vol. absolute ethanol were added, followed by centrifugation at 20,000 x g at 4°C. The precipitated DNA was resuspended in TE buffer (10 mM Tris pH 8.0, 1 mM EDTA in RNAse-free water) and 1 μL used as input material for the genotyping-PCR reaction: 1 x Q5 reaction buffer, 200 μM dNTPs, 200 μM forward and reverse primer (for locus-specific primer pairs see Table S2) and 0.3 U Q5 polymerase (NEB). The following PCR cycling conditions were used: 2 min at 98°C, 30 cycles of 30 s at 98°C, primer annealing for 20–30 s at 60°C–70°C, 30 s at 72°C, and a final extension of 5 min at 72°C. PCR products were purified via agarose gel electrophoresis and extracted using the Monarch DNA Gel Extraction Kit (NEB) before forwarding to Sanger sequencing (Eurofins Genomics). Sanger sequencing results were visualized and analyzed using Geneious Prime (2019.2.3, BioMatters).

### Microscopy

*In vivo* images of phenotypic-representative embryos were acquired 4 days post-injection with a Nikon SMZ18 equipped with Nikon DS-Ri1 and DS-Fi2 cameras. For imaging embryos were mounted into injection molds (1.5% (w/v) agarose in ERM). All images were processed using ImageJ.[116] ImageJ was used for image cropping and to adjust brightness and contrast settings.

### Wholemount *in situ* hybridization

Wholemount *in situ* hybridization using BCIP-NBT as chromogenic substrates was performed as previously described.[127] Stained embryos were mounted in 87% Glycerol on a microscope slide and imaged in a brightfield microscope (Nikon Eclipse 80i equipped with a pco.panda 4.2 camera). The RNA probes were synthesized from plasmids derived from an in-house cDNA library[111] ordered from Biocat or were cloned with the pGEM-T easy kit (Promega) using Mach1T1TM T1R phage resistant chemically competent *Escherichia coli* (Thermo Fisher Scientific) and transcribed with the T7 polymerase. In-house clones containing fragments covering the exons of ccdc141 (P13_C_01) and atg7 (P28E11) were used. For plasmid generation, PCR products derived from cDNA fragments of ppp3cca (ENSEMBL transcript ID: ENSORLT00000020175.2) and sptbn1 (ENSEMBL transcript ID: ENSORLT00000002513.2) were cloned. The template for the ryr2b probe was designed using the ENSEMBL transcript ID ENSORLT00000028007.1 and obtained from Biocat.

### Locus definition and fine mapping

The boundaries of the association regions were defined manually by visual inspection of the association profile and linkage disequilibrium structure to the lead SNP. The lead SNP was usually taken to be the variant with the smallest *p*-value in the locus, but excluding variants missing completely one homozygous state, or with impossible segregation patterns across the different F2 crosses.

### Candidate gene selection

Candidate genes (Table S2) for gene editing validation were selected within the association regions of each locus according to the presence of variants disrupting protein function that are also strongly linked to the lead variant for the locus, and previously reported cardiac effects associated with the gene. To check for previously reported cardiovascular effects for a gene, we used the phenotype tab of each gene's page on the ENSEMBL genome browser.[128] We also performed manual literature searches. To detect variants with a consequence on protein function that are in strong linkage with a lead variant for a locus, we used the *birneylab/varexplore* pipeline that we describe in the next section.

### The birneylab/varexplore pipeline

We reasoned that the causal mutation for an observed association signal may not be included in the set of genetic variants used for association testing. This may be because, for example, the causal variant is an insertion or deletion, while the set of variants that we used in the association testing were exclusively SNPs (due to limitations of the STITCH imputation software used).

To address this issue and detect putative causal variants that are not part of the tested set, we developed the *birneylab/varexplore* (https://github.com/birneylab/varexplore) Nextflow[129] pipeline, which we also make publicly available with associated documentation for other researchers to use. *birneylab/varexplore* can take advantage of linkage disequilibrium around a variant of interest to infer the presence of additional linked genetic variants. It requires in input aligned (low depth) sequencing reads for a set of samples, a file containing a set of variants of interest (for example lead SNPs from a GWAS analysis) and called genotypes for the same samples in *vcf* format (for example deriving from an imputation process). The pipeline extracts genotypes from the *vcf* file at the variants of interest, and clusters the samples according to their genotype at such variants (this is performed in parallel as separate processes for each variant). The sequencing reads are clumped together using *samtools*[125] according to the cluster to which each sample belongs (again, in parallel different clusterings are performed for different variants), and processed as meta-samples in downstream steps. These meta-sample reads are processed with *GATK* performing germline joint calling.[123] Variants in strong linkage disequilibrium with the grouping variants used, which may have been absent in the variant set used for the original GWAS, are expected to be called with different genotypes in the different meta-samples. On the contrary, variants that are not linked to the grouping variant will appear as heterozygous in all the meta-samples. The effects of the newly discovered variants are then predicted using ENSEMBL Variant Effect Predictor (VEP),[130] and the output is reformatted so that the variants and their effects can be directly loaded to the Integrative Genomics Viewer (IGV)[131] for manual exploration. Other outputs produced by the pipeline for further analysis are the meta-sample grouped reads in *cram* format, the meta-sample variant calls in *vcf* format, and the ENSEMBL VEP predictions.

To aid in filtering out variants that are not linked to the variants of interest, we also provide an R script,[132] *filter_variants.R*, within the pipeline repository that can parse the various outputs of the pipeline, and retain only variants that follow specific criteria in terms of genotype distribution across the meta-samples. For example, using this script it is possible to retain only variants that are called as opposite homozygous genotypes for the meta-sample corresponding to opposite homozygous genotypes at the grouping variant, and heterozygous for the meta-sample heterozygous at the grouping variant. More complex configurations are also possible.

For a more complete description of the pipeline usage and capabilities, the reader is encouraged to explore the *birneylab/varexplore* documentation at https://github.com/birneylab/varexplore.

## QUANTIFICATION AND STATISTICAL ANALYSIS

### Linear mixed model genetic association analysis

The association between genetic variants and the heart rate phenotype was analyzed using a mixed linear model, implemented in the *birneylab/flexlmm* pipeline.[117] The association analysis was run separately for the heart rate measurements taken at different

temperatures. Additional phenotypes consisting of the heart rate difference in the same samples across temperatures were also used in the association analysis. All the phenotypes were quantile normalized to a normal distribution. We used a random effect proportional to the genetic relatedness matrix of the samples, and fixed covariates corresponding to intercept, phenotyping plate ID and F2 cross ID. Significance was evaluated with a likelihood ratio test between a covariate-only model and a model that in addition to the covariates contained a term for the linear encoding of the genetic variant (homozygous reference encoded as 0, heterozygous encoded as 1, homozygous alternate encoded as 2) and a dominance term (heterozygous encoded as 1, any homozygous encoded as 0). Genome-wide significance thresholds were established as the minimum $p$-value obtained across all the SNPs tested over 100 permutations of the residuals (see Pierotti et al.[117] for methodological details).

The "Best model" column reported in Table 1 describes which specification of the statistical model reached the smallest $p$-value in a likelihood ratio test against a covariate-only model, also computed with the *birneylab/flexlmm* pipeline. In addition to the covariates, the G + E model contains the lead SNP for the locus additively encoded; the G + E + G×E model contains the SNP and its interaction with temperature; the G + E + D model includes the SNP and a dominance term; the G + E + D + G×E model includes the SNP, a dominance term, and the interaction of the SNP with temperature; the G + E + D + G×E + D×E model includes the SNP, a dominance term, and both of their interactions with temperature.

## Cross-specific QTL detection

To test the detectability of QTLs identified in the joint GWAS analysis within single crosses, we performed a post-hoc assessment (Figure S10). This was achieved by fitting cross-specific linear models testing the association between the lead SNP of each QTL and the phenotype that, in the discovery GWAS, achieved the highest significance for the respective locus. We evaluated $p$-values at a 5% False Discovery Rate across the full QTL by locus matrix.

## Detection of G×E and G×G effects

To detect the effect size and statistical significance of interaction terms, additional Nextflow pipelines were set up. These are available at https://github.com/birneylab/heart_rate_tmp_gxe and https://github.com/birneylab/heart_rate_tmp_pairwise_gxg. We performed both analyses uniquely on the marker SNPs for each locus. Differently than for the discovery GWAS, in this case we run the analysis jointly on the heart rate measured at different temperatures. The lead SNP of the *chr15_qtl* locus and its interaction with the temperature were included as a covariate in all the cases where *chr15_qtl* was not the target locus itself.

For the G×E analysis, we used a linear mixed model with 3 random effects covariance matrices. One covariance matrix was designed to correspond to a binary specification of the sample identity (1 for pairs of measurements relating to the same sample and 0 otherwise, to account for the repeated measurements of the same sample across temperatures). The second covariance matrix was derived from the genetic relatedness matrix and expanded to account for the repeated presence of the same samples at 3 different temperatures (it was computed as the Kronecker product of the GRM and a 3 × 3 matrix full of 1s). This second matrix modeled the relationship between samples across different temperatures as well as within the same temperature and thus served to estimate a genome-wide temperature-independent covariance. The third matrix was computed as the Kronecker product of the GRM and a 3 × 3 identity matrix. This last matrix served to encode the relatedness among samples within the same temperature, while the covariance across measurements taken at different temperatures was always set to 0. Thus, it estimated genetically determined covariance among samples within but not between temperature treatments. Together, the second and third matrix define a compound symmetry environment model where the variance and covariance among environments are constant (but possibly different from each other). The same fixed effects covariates used in the discovery GWAS were adopted. The GRMs were computed in a Leave-One-Chromosome-Out (LOCO) fashion, excluding the chromosome where the currently tested locus resided when calculating pairwise sample relatedness. The variance components were estimated using the *gaston* R package.[133] The variance components were estimated once per locus, using a fixed effect structure containing only an additive encoding for the locus and no dominance or G×E terms. The temperature was treated as a categorical variable, while a log-transformation of the heart rate was used as a phenotype for stabilizing the residual variance. Phenotyping batch and F2 cross were used as fixed-effect covariates. After estimation of the variance components, the random effects were regressed out in the same way described in Pierotti et al.[117] Linear models were then fitted to the decorrelated data to estimate effect sizes. To calculate the variance explained by the additive genetic (G), dominance (D), gene-by-environment (G×E), and dominance-by-environment (D×E) terms (Figure 3D), an ANOVA model (including covariates) was fitted on the decorrelated data. The sum of squares for each term was then divided by the total sum of squares for the model. To calculate the variance explained as a proportion of the overall variance explained by genetic terms (Figure 3E), the sum of squares for each term was divided by the total sum of squares for the G, D, G×E, and D×E terms. To assess statistical significance of interaction terms (Figure 3C), likelihood ratio tests were performed. The $p$-values were Bonferroni-corrected across QTLs and interaction modes (Dominance, G×E, D×E; number of tests: 48) and assessed at a 0.05 threshold.

The G×G analysis (Figure 4A) was performed similarly to the G×E analysis, using the same structure for the random effects. In addition to the covariates used in the G×E analysis, the marginal effect of the interacting loci tested and their marginal G×E effect were included as covariates. The GRMs were computed excluding 2 chromosomes at a time (for the 2 interacting loci modeled). Significance was determined via likelihood ratio tests and Bonferroni-corrected across QTLs and interaction modes (G×G, G×G×E) to account for the number of tests performed ($n$ = 240).

Our experimental design was adequately powered to detect GxG interactions. In the ideal scenario of 2 interacting G×G loci, both at 50% allele frequency in the same F2 cross and no linkage, one would expect 6.25% of samples in each of the minority classes (double homozygous genotypes). For our average F2 cross of 220 samples, this translates to approximately 14 samples in the smallest genotype class.

### Calculation of variance explained by single loci

The variance explained estimates reported in Figures 3D and 3E and Data S7 were derived from the locus-specific linear model fits used in the G×E analysis described in the previous section. For each locus, variance explained was calculated as the partial $\eta^2$ for the genetic terms—defined as the ratio of the sum of squares attributable to the genetic effects (including interactions) to the residual sum of squares from the ANOVA.

### Statistical analysis of gene editing results

Gene editing validation results (Figure 5B) were modeled with a mixed effect model using the *lmerTest* R package.[134] Random intercepts and random slopes with respect to temperature were included to account for repeated measurements of the same samples across different temperatures. The heart rate phenotype was log-transformed to stabilize residual variance, and temperature was also log-transformed to linearize its relationship with the transformed heart rate (as in the simulations described in the next section, we used $log(t-6)$, where $t$ is the temperature in degrees Celsius). All p-values were FDR-corrected.

For plotting only (but not for statistical analysis), phenotypes were mean centered based on the mean of the control samples, and temperature was shown on the original scale.

Data points were considered strong outliers and excluded from the analysis following Tukey's definition[135] if they fell outside the range:

$$[Q_1 - k(Q_3 - Q_1), Q_3 + k(Q_3 - Q_1)]$$

With $Q_1$ and $Q_3$ the first and third quartiles respectively. The value of $k$ was conservatively set to 4, marking only exceptionally strong outliers (by comparison, $k = 1.5$ defines standard Tukey outliers, and $k = 3$ marks points that are "far out").

The data presented in Figure 4D were analyzed similarly. Separate mixed models were fit, stratified by background medaka strain (with results shown in each subplot), and the resulting p-values were FDR-corrected. A joint mixed model was also fit to the data, treating strain and its interaction with gene editing status as categorical fixed effects (results shown in the top right of Figure 4D).

### Simulation of the effects of model misspecification

To assess the detectability of QTLs with effect sizes similar to those observed in the medaka dataset under varying levels of model misspecification, we developed a simulation framework. The corresponding simulation code is available as a Nextflow pipeline at: https://github.com/birneylab/heart_rate_tmp_simulation_gxg.

For each medaka QTL, we simulated the same locus at a range of allele frequencies and sample sizes, with 100 replicates per combination. QTL genotypes were generated by sampling from a binomial distribution with two trials and success probability equal to the target allele frequency. Environmental data (temperature) were sampled with replacement from outdoor water temperature measurements collected in the MIKK panel's native geographic region in Japan (courtesy of Dr. Nakayama, Nagoya University and used also in Nakayama et al.[136]). These measurements were recorded from the middle layer of a 60 L water container every hour over two years (1 October 2015–15 October 2017). We restricted our sampling to values between 21°C and 35°C during the medaka reproductive season, avoiding extrapolation beyond the range directly observed in the GWAS.

Except for *chr15_qtl* and *chr21_qtl* (discussed below), we fitted a linear mixed model to the medaka GWAS data for each locus, using the same model framework described previously in the context of estimating G×E and dominance effects. Effect sizes were estimated under two models: a G + E + D + G×E + D×E model, and an extended model that also included G×G and G×G×E terms (labeled "GxG + GxGxE no" and "GxG + GxGxE yes", respectively, in Figure 6). G×G and G×G×E interactions were estimated between each focal locus and the lead SNPs at *chr15_qtl* and *chr21_qtl*, for which causal variants had been identified and their frequencies in the wild determined. Consequently, the *chr15_qtl* and *chr21_qtl* loci themselves were not used as focal loci in simulations. The *chr15_qtl* and *chr21_qtl* loci were always simulated at their wild allele frequency under a binomial distribution. The marginal effect of *chr15_qtl* and *chr21_qtl* and their interaction with temperature were always included as a fixed effect covariates, both in the "GxG and GxGxE yes" and "GxG and GxGxE no" models.

Unlike the approach used for G×E and dominance effect detection described earlier, here we treated temperature as a continuous variable to support simulations across a broader, realistic range of values —not just 21, 28, and 35°C. We log-transformed the heart rate phenotype to stabilize residual variance. To linearize the relationship between temperature and log-transformed heart rate in the 21°C–35°C range, we also applied a log-based transformation to the temperature (we used $log(t-6)$ where $t$ is the temperature in degrees Celsius). These transformations were consistently applied to both the medaka data (for effect size estimation) and the simulated data (for discovery).

Synthetic phenotypes were generated by summing the products of the simulated values of each term (intercept, G, E, D, G×E, D×E, G×G, G×G×E) and the effect sizes estimated from medaka. For the "GxG and GxGxE no" column in Figure 6 the epistatic terms were omitted. Gaussian noise with variance equal to the residual variance of the corresponding medaka locus was then added.

We did not include covariates or random effects in the simulations, as these were specific to our experimental setup and population.

We conducted QTL discovery on the simulated data using models of increasing complexity. In each case, we performed a likelihood ratio test comparing the full model to a reduced model that included only temperature, intercept, *chr15_qtl* and *chr21_qtl*, and their interactions with temperature. In the case of the G-only model (without environment), the comparison was made against a model with just the intercept, *chr15_qtl* and *chr21_qtl*. To evaluate the impact of environmental noise, we introduced Gaussian noise to the temperature variable such that noise accounted for 1%, 10%, or 50% of the total temperature variance. The "generating model" row in Figure 6 refers to simulations evaluated under models that either included (for the "GxG and GxGxE yes" column) or excluded (for the "GxG and GxGxE no" column) the G×G and G×G×E terms.

We also evaluated the effect of using tagging rather than causal SNPs by replacing the causal variant used to generate the phenotype with an SNP with known linkage (e.g., $r^2 = 0.9$) to the causal variant, and recomputing all interaction terms accordingly (Figure 6, "causal variant" vs. "$r^2 = 0.9$" columns).

Each simulation condition (combination of QTL, sample size, allele frequency, epistasis model, and tagging level) was repeated 100 times. For each replicate, we identified the minimum sample size at which genome-wide significance ($p < 5 \times 10^-$) was achieved. Median minimum sample sizes across replicates are shown in gray in Figure 6, with the median across QTLs plotted as a red line.

In Figure S6, we present QTLs ordered by their degree of non-additivity. We quantified non-additivity using a dominance ratio:

$$d / g = (\beta_D + \beta_{D \times E} \bar{E}) / (\beta_G + \beta_{G \times E} \bar{E}))$$

where $\beta_G$, $\beta_{G \times E}$, $\beta_D$, and $\beta_{D \times E}$ are the effect sizes for a given QTL, and $\bar{E}$ is the mean environmental value. This analysis used estimates from the "GxG and GxGxE no" models. The absolute value of the dominance ratio represents the degree of non-additivity; loci with a positive ratio are classified as dominant, and those with a negative ratio as recessive. In Figure S6, loci are color-coded according to dominance/recessiveness. To visualize the reciprocal condition, we also plot each locus with the reference and alternate alleles swapped.

In Figure S8 we instead show the sample size required to discover the non-additive effects, specifically dominance (D) and GxE interactions. In this analysis, likelihood ratio tests were conducted against a null model that included the additive effect of the locus, the environment, *chr15_qtl*, *chr21_qtl*, and the intercept.

## Construction of gene trees for confirmed genes

The trees shown in Figure S9 were built with *iqtree2*[137] based on protein multiple sequence alignment performed with *t-coffee*,[138] both run with default parameters. Visualizations were produced with *iTOL*,[139] and homologues were retrieved from *OrthoDB*.[140] The set of species included in each tree varied depending on the availability of annotated homologues, but human, mouse, chicken, and medaka sequences were always included. The brownbanded bamboo shark (*Chiloscyllium punctatum*) or the roundworm (*Caenorhabditis elegans*) was used as an outgroup to root each tree. Very short or poorly aligned sequences were excluded from the analysis.

## Statistical tests for reciprocal cross effect

To test whether the maternal versus paternal origin of the founding strain influenced QTL effects, we focused on reciprocal F2 crosses (72-2×139-4 and 139-4×72-2; $n = 302$), in which the parental strains were swapped in maternal and paternal roles. We further restricted the analysis to QTLs for which each reciprocal cross included at least 10 samples in the least frequent genotype. Analyses were stratified by temperature treatment and QTL, using an inverse quantile normalization of the heart rate phenotype. For each combination, we fitted a null linear model including phenotyping plate as a covariate, as well as the main effects of QTL and reciprocal cross. This was compared via Likelihood Ratio Tests (LRTs) to an alternative model that additionally included the QTL × reciprocal cross interaction. Resulting *p*-values were corrected using both FDR and Bonferroni adjustments (Data S9).

Separately, we tested for a marginal effect of reciprocal cross status by comparing a model with only phenotyping plate as a covariate to one that included both plate and reciprocal cross (without any genetic terms). As before, significance was assessed using LRTs and corrected with a Bonferroni correction.

We found no evidence of interaction between the reciprocal cross and genotype at any of the 16 QTLs after Bonferroni correction (Data S9). On the contrary, when we looked into the marginal effect of the reciprocal cross status independent of genetics, we were able to detect significant effects on heart rate after Bonferroni correction at all the temperature treatments (21°C: $p = 8.06 *10^{-3}$; 28°C: $p = 7.29*10^{-8}$; 35°C: $p = 5.18*10^{-4}$).

## Human height UK Biobank simulation example

To confirm that the statistical simulation approach used in Figure 6 is appropriate under the stated assumptions, we applied an analogous procedure to a human GWAS dataset from the Neale Lab UK Biobank resource (http://www.nealelab.is/uk-biobank/). Specifically, we downloaded the summary statistics for standing height from:

https://broad-ukb-sumstats-us-east-1.s3.amazonaws.com/round2/additive-tsvs/50_raw.gwas.imputed_v3.both_sexes.tsv.bgz -O 50_raw.gwas.imputed_v3.both_sexes.tsv.bgz.

From the reported minor allele frequency ($p$) and standard error ($SE_\beta$), we computed the residual variance for each marker under the assumption of Hardy–Weinberg equilibrium as $\sigma_\varepsilon^2 = SE_\beta^2 * n * \sigma_x^2$, where the variance of the marker is $\sigma_x^2 = 2 * p * (1 - p)$ and $n$ is the sample size.

For each marker, we then generated a synthetic genotype vector $x_{sym}$ by sampling from a binomial distribution with two trials and success probability $p$. We sampled a residual vector $\varepsilon_{sym}$ from a normal distribution with 0 mean and variance $\sigma_\varepsilon^2$, and constructed a synthetic phenotype $y_{sym} = \beta x_{sym} + \varepsilon_{sym}$.

After fitting a linear model to the simulated data, we obtained Wald test $p$-values and compared them with those observed in the GWAS (Figure S5). The code for this simulation is available at https://github.com/birneylab/heart_rate_tmp_simulation_gxg/tree/main/ukb_example.

The simulated and observed $p$-values are strongly correlated and show no systematic bias, indicating that our approach correctly recovers the expected level of significance for the additive genetic term. Since the simulation is parameterized only by effect sizes, residual variance, and the distribution of environmental factors, its relevance depends on how accurately these parameters reflect reality. If the parameters estimated in medaka differ from those governing the trait of interest, the results should therefore be interpreted with caution.

### In addition, we provide the following list of assumptions and caveats

- Genetic effect sizes estimated in medaka are assumed to accurately represent those of the trait of interest.
- The residual error estimated in medaka is assumed to reflect the residual error of the trait of interest.
- Variants linked to the focal variant (e.g., compound heterozygotes) were not modeled, although imperfect tagging of causal variants by observed markers was included.
- Only a single environmental factor was considered; multiple or potentially correlated exposures were not modeled.
- Confounding arising from correlations between population structure and environmental exposures was not modeled.

