## [Document S2. Transparent peer review records for Welz et al. · Cell Genomics]

Discovery and characterization of gene by environment and epistatic genetic effects in a vertebrate model

Bettina Welz, Saul Pierotti, Tomas Fitzgerald, Thomas Thumberger, Risa Suzuki, Philip Watson, Jana Fuss, Tiago Cordeiro da Trindade, Fanny Defranoux, Marcio Ferreira, Kiyoshi Naruse, Felix Loosli, Jakob Gierten, Joachim Wittbrodt, Ewan Birney

Summary

Initial submission: Received : Jun 18, 2025

Scientific editor: Sara Rohban

First round of review: Number of reviewers: 3
Revision invited : Aug 11, 2025
Revision received : Oct 07, 2025

Second round of review: Number of reviewers: 1
Accepted : Jan 14, 2026

Data freely available: YES

Code freely available: YES

This transparent peer review record is not systematically proofread, type-set, or edited. Special characters, formatting, and equations may fail to render properly. Standard procedural text within the editor's letters has been deleted for the sake of brevity, but all official correspondence specific to the manuscript has been preserved.

Referees' reports, first round of review

Reviewer #1:

Welz and colleagues have submitted a manuscript where they explore gene by environment interactions in a controlled way using inbred strains of medaka, heartrate measurements, and regulated temperature changes. Because each of the inbred medaka lines has been fully sequenced, it was possible to select strains with significant heart rate divergence from the average, either heartrates hyper-responsive to temperature or stable differences in rate, and perform genetic crosses to map the QTLs. These crosses also allowed the investigators to determine the mode of genetic inheritance, e.g. dominant, recessive, GxG and/or GxE. The researchers were able to identify 16 qtl regions and identified the causative variant in 4 of them. Each of the 4 was confirmed by gene editing to mimic the existing variant and observed phenotype. Using this data, they were able to model sample sizes in human data needed to identify causative variants based on different inheritance models. Notably, the simulations suggest that genetic or environmental interactions would not be detectable in most human GWAS studies as they are underpowered. I believe these findings are of interest to a broad range of geneticists and bioinformaticians.

Comments:

This is a rare instance where I do not have any significant critiques to the approaches presented in manuscript. The experiments as designed, the analysis of the results, and the validation studies are carefully done and reasonable. Other reviewers will have to evaluate the modeling predictions as it is beyond my expertise. I only have minor comments that should be addressed in the text:

- 1) The authors should discuss the limitations of mapping more thoroughly. They were successful in identifying the causal variant in 4 cases because there were snv in coding regions of genes where it was already known the genes impacted heart function in some way. Different approaches will be necessary to identify the majority of the qtl's and the researchers face the same challenges that plague most of the GWAS data.
- 2) The validation studies focused on injected larvae that looked "normal" but the crispants and editants showed higher than expected heart morphological phenotypes given that the identified snv's appear to generate truncated alleles but do not have visible phenotypes. Were abnormal embryos tossed out before the initial screening? Are there reasons to believe the edited alleles are more severe than the identified variants? Some additional discussion on this issue would be useful.

Reviewer #2:

This study investigates the genetic architecture of heart rate in medaka fish (*Oryzias latipes*) under controlled temperature conditions, leveraging the Medaka Inbred Kiyosu-Karlsruhe (MIKK) panel to dissect gene-by-environment (G×E) and epistatic (G×G) interactions. Overall, this is an impressive study that represents an elegant approach and developed wild-derived medaka strains as model to study G×E interactions with controlled environmental manipulation. The identification and characterization of multiple types of genetic effects (additive, dominance, G×E, G×G, G×G×E) provides a nuanced view of complex trait architecture. I have few comments hope the authors can address before the manuscript can be accepted for publication.

1. The manuscript presents 16 QTLs identified from 11 crosses among 8 MIKK strains but did not report which QTLs were detected in which specific crosses. Do any QTLs appear in multiple independent crosses, which would constitute high-confidence loci? If most QTLs are cross-specific (detected in only one cross), this raises concerns about false discovery rates and the robustness of the findings. The authors should provide a cross-discovery matrix showing which QTLs were detected in each cross, calculate replication rates, and discuss the biological and methodological implications. If replication is low, they should address potential explanations including population structure artifacts, epistatic interactions, or power limitations, and acknowledge how this affects the reliability of their conclusions.

2. The methodology for testing G×E, G×G, and G×G×E interactions is conceptually interesting but inadequately described. The authors should summarize the statistical principles underlying these interaction tests in the main text, not just hidden them to supplementary methods. More critically, it is unclear how multiple testing correction was applied across the different types of analyses. Were corrections applied separately for each interaction type, or jointly across all tests? Given the large number of comparisons, the multiple testing burden is substantial. The authors should provide explicit details about their correction strategy and justify their approach, as inadequate correction could lead to inflated Type I error rates.

3. While the strain-dependent response to *ccdc141* editing is statistically significant and biologically interesting, it does not constitute rigorous validation of epistasis. The experiment tests *ccdc141* editing across different inbred strains with completely different genetic backgrounds, making it impossible to attribute the strain-dependent effects to specific epistatic interactions rather than confounding factors such as baseline heart rate differences or other genetic modifiers. The term "overall genetic background" is too broad and undefined to constitute proper epistatic testing. True epistatic validation requires testing specific, defined genetic interactions between identified loci, not general background effects. Thus this experiment provides suggestive evidence for genetic background modulation but falls short of the rigorous epistasis validation claimed by the authors.

4. The authors engage in circular reasoning when claiming that their validation success "is unlikely to be attributable to random chance in selecting genes with heart rate effects." Since they pre-selected genes based on cardiac function annotations and predicted severe

consequences, then used this targeted selection as evidence of their method's validity. And comparing their targeted approach to the Hammouda et al. (2021) study showing 20% success with random gene selection is misleading, as it contrasts a biased selection strategy with an unbiased one. The authors should either acknowledge the selection bias and remove claims about statistical significance of their success rate, or conduct proper validation using unbiased gene selection.

5. The simulation section addressing human GWAS applicability suffers from fundamental limitations that undermine its conclusions. The simulations assume a relatively simple genetic model based on medaka QTLs, which cannot adequately reflect human complex trait architecture where hundreds to thousands of causal variants typically contribute to phenotypic variation. Critical parameters differ substantially between the medaka system and human populations, including linkage disequilibrium patterns, allele frequency distributions, and population structure. Most problematically, the simulations treat environmental factors as precisely measured continuous variables, which rarely holds in human studies where environmental exposures are typically measured with substantial error and temporal variation. Given these limitations, the conclusions about human GWAS design are not well-supported and may mislead readers. I recommend removing this entire section and focusing the manuscript on the valuable medaka-specific findings, which stand on their own merit without requiring extrapolation to human genetics.

Reviewer #3:

Review of "Discovery and characterization of gene by environment and epistatic genetic effects in a vertebrate model." In this manuscript, Welz et al. phenotype over 80 inbred medaka (*Oryzias latipes*) lines for embryonic heart rate, perform 11 sets of crosses with 8 phenotypically divergent lines, and use low-coverage WGS on F2 offspring and to identify QTLs. Commendably, the authors also experimentally validate the QTLs with gene editing (e.g., CRISPR) and perform simulations to determine the effect of epistasis on locus discovery in human GWAS studies. Overall, the authors performed a very large amount of well-designed work, the analyses and results are compelling, and the manuscript is very clearly written. A number of comments follow, the majority of which should be straight forward to address.

Comments:

The authors identify ~ 16 QTL using methods and thresholds similar to those used in human studies where conservative thresholds are required for obvious reasons. However, 16QTL is not very many and limits the ability to account for epistatic interactions. I appreciate that there is one locus of very large effect and also appreciate the approach the authors use as re-running the analysis with that locus as a covariate. Nevertheless, I think it would be informative to use a substantially less stringent threshold(s), identify many more candidate QTLs, and re-run the GxG and GxGxE associations. This information could be largely relegated to SI, with a few additional lines to the main text summarizing what was found.

I am a bit skeptical of imputation methods on F2 individuals. Imputation works well in highly inbred populations, but the F1 parents are highly heterozygous. I appreciate that there is a published paper documenting this method, but additional details to reduce this skepticism in future readers would be useful (were the same strains used, were any sequenced to higher depth.). Could the imputation be responsible for the fact that almost the entirety of chromosome 15 has significant p-values for QTL analyses? If not, please make it clear why.

Why was only a single reciprocal cross performed? Are the sex-determining loci/regions known in medaka? Is sex genetically determined? I

Line comments:

Unfortunately, neither line number nor page numbers were provided in my review copy. As such the line comments follow by section and the first 3-5 words that the sentence starts with (can search to find line).

Abstract:

"showing that apparent additivity..." could this result be made more clear - "apparent additivity in human GWAS" is vague; also what are the implications for phenotypic prediction?

Introduction:

"Nonetheless, even in human genetics" - I thought this sentence was going to say something else entirely. Please revise the first few words for clarity.

"...such as dominance and epistatic effects (Currant)" - a bit vague; could specific dominance and epistatic effects be mentioned?

The paragraph about mice is okay, but probably not needed

"It's high fecundity, ex-utero..." - it was not immediately clear why ex-utero was an advantage - but probably is a big advantage for the phenotype of choice. Consider making more clear

"with recorded ecological limits (35C)" - need a reference for this statement

Somewhere in the legend of Figure 1, can you make it clear that the data are heuristic (i.e., not real) - I initially thought the data in panel 1 were empirical data.

Results:

"...we observed sub-clustering of F2..." - was the sub-clustering random or driven by cross-type? Those details should be added here.

I really like Figure 2 - pretty amazing data! One minor comment, the double headed vs single headed arrows in panel C are confusing and not explained anywhere in the legend. Also, the reciprocal cross should be identified in the legend.

"On the contrary,..." I did not follow this reasoning - could this be explained more clearly (also may help resolve 3rd main comment from above)?

Figure 3: Why were only 100 permutations performed? Could 1000 permutations be performed, for example.

"we observed that 21 out of 120 possible QTL pairs are significant...GxG.." - Was there any correlation between physical distance and GXG or GXGXE loci? That is, were pairs closer together in the genome more likely to be GXG or were they all on 16 separate chromosomes?

Figure 4 c: It would be useful to know which temperature treatments (individually) were significant after accounting for multiple comparisons. More importantly, how much is the overall significant result driven by a single or only a handful of temperature treatments?

"; in each of the four cases the effect validated by..." - wording is strange here - "the effect was consistent" - do you mean the predicted vs. measured shift in phenotype or just the fact that loci had an effect? Please clarify.

"respect to two loci- chr15_qtl and chr21_qtl..." why these two instead of those shown in Figure 5?

Choice of MAF filter (and other filters) both here and for GWAS are likely important. See <https://doi.org/10.1038/s41576-024-00738-6>

Figure 6 could be a little more clear. Rather than simple yes vs no, could you also color code and include epistasis vs no epistasis or something along those lines?

Figure 6 (and discussion): there is a big difference between locus discovery and phenotypic prediction; it seems epistasis would have a large effect on the latter and, as you show, a small effect on the former. This distinction could be added to the abstract and talked about more in the Discussion.

Discussion:

any thoughts on higher order epistatic interactions (≥ 3 QTLs/loci/genes) would be interesting to add.

great paper overall!

Authors' response to the first round of review

Reviewer #1

Welz and colleagues have submitted a manuscript where they explore gene by environment interactions in a controlled way using inbred strains of medaka, heartrate measurements, and regulated temperature changes. Because each of the inbred medaka lines has been fully sequenced, it was possible to select strains with significant heart rate divergence from the average, either heartrates hyper-responsive to temperature or stable differences in rate, and perform genetic crosses to map the QTLs. These crosses also allowed the investigators to determine the mode of genetic inheritance, e.g. dominant, recessive, GxG and/or GxE. The researchers were able to identify 16 qtl regions and identified the causative variant in 4 of them. Each of the 4 was confirmed by gene editing to mimic the existing variant and observed phenotype. Using this data, they were able to model sample sizes in human data needed to identify causative variants based on different inheritance models. Notably, the simulations suggest that genetic or environmental interactions would not be detectable in most human GWAS studies as they are underpowered. I believe these findings are of interest to a broad range of geneticists and bioinformaticians.

Comments:

This is a rare instance where I do not have any significant critiques to the approaches presented in manuscript. The experiments as designed, the analysis of the results, and the validation studies are carefully done and reasonable. Other reviewers will have to evaluate the modeling predictions as it is beyond my expertise. I only have minor comments that should be addressed in the text:

We thank the reviewer for the positive feedback on our work.

1) The authors should discuss the limitations of mapping more thoroughly. They were successful in identifying the causal variant in 4 cases because there were snv in coding regions of genes where it was already known the genes impacted heart function in some way. Different approaches will be necessary to identify the majority of the qtl's and the researchers face the same challenges that plague most of the GWAS data.

We thank the reviewer for this comment. As noted, the mapping population used in this study is not well suited for high-resolution fine mapping due to the extensive linkage disequilibrium inherent to the F2 design. Alternative strategies would therefore be required to fine-map additional loci. In line with the reviewer's suggestion, we have expanded the discussion to further emphasise these fine-mapping limitations.

“As expected, fine mapping in our F2 population was challenging due to the limited number of recombination events, which resulted in strong correlations among neighbouring genetic variants. Consequently, causal variants could mostly be identified in QTLs where severe loss-of-function mutations were detected (ryr2b, ccdc141), or where genes with relevance to heart function were present (ryr2b, ccdc141, sptbn1). For the remaining QTLs, pinpointing candidate genes will require additional mapping populations with reduced linkage disequilibrium. Potential approaches include using outbred wild samples, extending the breeding scheme beyond the second generation, backcrosses or increasing sample sizes. Such strategies may also enable the detection of variants exerting more subtle regulatory effects on gene function.”

2) The validation studies focused on injected larvae that looked "normal" but the crispants and editants showed higher than expected heart morphological phenotypes given that the identified snv's appear to generate truncated alleles but do not have visible phenotypes. Were abnormal embryos tossed out before the initial screening? Are there reasons to believe the edited alleles are more severe than the identified variants? Some additional discussion on this issue would be useful.

We thank the reviewer for the questions and appreciate the opportunity for clarification. We would like to specify that during the initial phenotyping of the MIKK panel strains and heart rate acquisition of F2 embryos, no filtering or removal of abnormal embryos was performed in order to ensure an unbiased analysis and maintain the full range of phenotypic diversity. However, prior to heart rate acquisition, we did verify that the embryos were gross morphologically normal and at the correct developmental stage. The reviewer is correct that editing the candidate genes in the embryos led to an increased proportion of embryos with cardiac phenotypes compared to the MIKK panel strains and F2 embryos respectively. Differences in the phenotype severity between the injected mosaic F0 generation and a mutant allele carrier have been reported in other studies, and could be due to genetic compensation (PMID: 37223518). Genetic compensation, which occurs when genetic changes are buffered by compensatory mechanisms, can reduce the severity of phenotypes in animals with fixed alleles. However, this compensation often does not occur in the injected F0 generation of animals with mosaic genotypes, resulting in more pronounced phenotypes and variable penetrance (PMID: 37223518). When applying the CRISPR/Cas9 to introduce frame shift mutations, different insertion and deletions are generated. Thus, this approach can result in a spectrum of alleles that translate into the observed phenotype and by that might differ in expressivity compared to the isogenic variant-carriers. Nevertheless, several studies have demonstrated the utility of base editing and the CRISPR targeting approach for a fast, efficient, and reliable validation and characterization of gene function in the F0 generation (PMID: 25867848, 29974860, 35373735, 37584388).

“To ensure an unbiased analysis and maintain the full range of phenotypic diversity no filtering or selection of embryos besides their developmental stage was performed.”

Reviewer #2

This study investigates the genetic architecture of heart rate in medaka fish (*Oryzias latipes*) under controlled temperature conditions, leveraging the Medaka Inbred Kiyosu-Karlsruhe (MIKK) panel to dissect gene-by-environment (G×E) and epistatic (G×G) interactions. Overall, this is an impressive study that represents an elegant approach and developed wild-derived medaka strains as model to study G×E interactions with controlled environmental manipulation. The identification and characterization of multiple types of genetic effects (additive, dominance, G×E, G×G, G×G×E) provides a nuanced view of complex trait architecture. I have few comments hope the authors can address before the manuscript can be accepted for publication.

We thank the reviewer for the positive feedback on our manuscript.

1. The manuscript presents 16 QTLs identified from 11 crosses among 8 MIKK strains but did not report which QTLs were detected in which specific crosses. Do any QTLs appear in multiple independent crosses, which would constitute high-confidence loci? If most QTLs are cross-specific (detected in only one cross), this raises concerns about false discovery rates and the robustness of the findings. The authors should provide a cross-discovery matrix showing which QTLs were detected in each cross, calculate replication rates, and discuss the biological and methodological implications. If replication is low, they should address potential explanations including population structure artifacts, epistatic interactions, or power limitations, and acknowledge how this affects the reliability of their conclusions.

We thank the reviewer for this observation and have computed a QTL-by-cross table as suggested (**Figure S9**). In our original analysis, the 11 F2 crosses were modelled jointly using mixed-effect models to account for uneven relatedness among samples. This approach does not allow direct testing of whether a given QTL is detectable in a specific cross within the original association framework. Moreover, a single cross is often underpowered to reach genome-wide significance for QTLs that can nonetheless be detected when data are aggregated across multiple crosses.

To address the reviewer's point, we performed QTL- and cross-specific association tests restricted to the lead variants of the 16 identified QTLs. For each QTL, we tested the phenotype with the strongest association in the original GWAS and applied a Benjamini–Hochberg correction to the full cross-by-QTL association matrix at a 5% False Discovery Rate. We observe a high degree of replication of our association signals, with all the QTLs except for one being detected in more than one F2 cross. The absence of replication in certain crosses is expected and may reflect: (1) the causal allele not being polymorphic in the founder individuals; (2) differences in linkage disequilibrium between the true causal variant and the lead SNP across crosses; or (3) insufficient power when testing within a single cross compared with the full GWAS population comprising 11 crosses.

Even when the same founding strain is used, different crosses can show distinct association patterns due to the incomplete homozygosity of the founder strains. The causal allele for a locus may segregate differently across individuals, or may display different linkage to the tested lead SNP. Additional differences in the association profiles of the same locus across F2 crosses may also arise from epistatic interactions.

2. The methodology for testing G×E, G×G, and G×G×E interactions is conceptually interesting but inadequately described. The authors should summarize the statistical principles underlying these interaction tests in the main text, not just hidden them to supplementary methods. More critically, it is unclear how multiple testing correction was applied across the different types of analyses. Were corrections applied separately for each interaction type, or jointly across all tests? Given the large number of comparisons, the

multiple testing burden is substantial. The authors should provide explicit details about their correction strategy and justify their approach, as inadequate correction could lead to inflated Type I error rates.

We thank the reviewer for this observation. Each figure has its own multiple-testing correction, so significance should be interpreted on a per-figure basis rather than across the entire study or for individual interaction types. Specifically, Figure 3C was corrected jointly for multiple testing across all loci and interaction modes (Dominance, G×E, D×E), while Figure 5A was corrected across loci and interaction modes (G×G, G×G×E). The two figures, however, were corrected independently of one another.

A full description of the statistical methodology for the interaction tests is provided in the STAR Methods section, which we have slightly revised to clarify the exact correction procedure and number of tests performed.

For G×E, Dominance, and D×E:

“To assess statistical significance of interaction terms (Figure 3C), likelihood ratio tests were performed. The p-values were Bonferroni-corrected across QTLs and interaction modes (Dominance, G×E, D×E; number of tests: 48) and assessed at a 0.05 threshold.”

For G×G and G×G×E:

“Significance was determined via likelihood ratio tests and Bonferroni-corrected across QTLs and interaction modes (G×G, G×G×E) to account for the number of tests performed (n = 240).”

In this latter case, the total number of tests and correction factor had previously been incorrectly reported as “120” instead of “240”; we have corrected this accordingly.

As suggested by the reviewer, we also added a brief clarification in the *Results* section describing our methodology and multiple-testing correction strategy:

For G×E, Dominance, and D×E:

“To quantitatively assess interaction effects and further characterise the genetic architecture of the heart rate phenotype, we evaluated both the significance and the proportion of variance explained by different interaction terms in our dataset. For each QTL, heart rate measurements at 21, 28, and 35 °C were analysed jointly, and likelihood ratio tests were performed to compare linear models with or without the interaction term of interest. Relatedness and measurement temperature were accounted for using mixed-models (see STAR Methods for details). Multiple testing was controlled with a Bonferroni correction based on the total number of tests across QTLs and interaction modes (n = 48; 16 QTLs and 3 interaction modes).”

For G×G and G×G×E:

“To control for the multiple-testing burden, we restricted our analysis to interactions among the 16 QTLs identified in this study. We applied the same statistical framework used for the Dominance and G×E tests described above (see STAR Methods for details) and implemented a Bonferroni correction across QTLs and interaction modes (n = 240; 120 possible QTL pairs × 2 interaction modes). This correction was performed independently of the one applied to the Dominance and G×E tests in Figure 3C.”

3. While the strain-dependent response to *ccdc141* editing is statistically significant and biologically interesting, it does not constitute rigorous validation of epistasis. The experiment tests *ccdc141* editing across different inbred strains with completely different genetic backgrounds, making it impossible to attribute the strain-dependent effects to specific epistatic interactions rather than confounding factors such as baseline heart rate differences or other genetic modifiers. The term "overall genetic background" is too broad and undefined to constitute proper epistatic testing. True epistatic validation requires testing specific, defined genetic interactions between identified loci, not general background effects. Thus this experiment provides suggestive evidence for genetic background modulation but falls short of the rigorous epistasis validation claimed by the authors.

We thank the reviewer for raising this concern.

Baseline heart rate variability cannot constitute a confounding factor in this case because it is accounted for as "medaka strain" covariate in the linear model. The test that we performed is a likelihood ratio test among a model including the strain and gene editing status as covariate and a model including, in addition to those terms, their interaction.

As the reviewer suggests, our result cannot validate the presence of specific epistatic interactions among QTLs. Indeed, we don't claim that our results validate any specific instance of epistasis. Instead, we claim that the presence of a differential effect of gene editing across strains is evidence of epistasis between the edited gene and other, unidentified, genetic variants that differ across the tested strains. Thus, we confirmed that epistasis with the editing of *ccdc141* is detectable, even though with this experiment we cannot identify the interacting partners.

We modified the relevant section to clarify this point:

*"The strain-dependence of the genetic effect is statistically significant ($p = 8.98 \times 10^{-3}$) providing evidence for a G×G interaction between the *ccdc141* edit and other, unidentified genetic variants that differ across strains. However, it should be noted that this experiment does not validate the presence of any specific epistatic pair. Rather, it demonstrates that the effect of genetically altering *ccdc141* is modulated by the strain's genetic background."*

4. The authors engage in circular reasoning when claiming that their validation success "is unlikely to be attributable to random chance in selecting genes with heart rate effects." Since they pre-selected genes based on cardiac function annotations and predicted severe consequences, then used this targeted selection as evidence of their method's validity. And comparing their targeted approach to the Hammouda et al. (2021) study showing 20% success with random gene selection is misleading, as it contrasts a biased selection strategy with an unbiased one. The authors should either acknowledge the selection bias and remove claims about statistical significance of their success rate, or conduct proper validation using unbiased gene selection.

We thank the reviewer for this observation. It is correct that our selection strategy is biased — favouring genes with prior heart-related annotations and those located within QTLs — and we contrast it with an unbiased random gene selection approach. However, this bias is intentional: our aim with this comparison is to demonstrate that a targeted strategy based on GWAS and previous knowledge for candidate gene validation is more effective than random selection.

The higher success rate we observe can be attributed to two factors: (1) the preferential selection of genes with previously reported links to heart function in other species, and (2) the restriction of our search to genomic regions showing a QTL signal.

To clarify that we cannot attribute the success rate solely to the presence of a QTL signal, but rather to the overall selection strategy — which includes prioritising genes with known associations to heart rate in other species — we have revised the discussion section accordingly:

“The success of our CRISPR validation experiments is unlikely to be attributable to random chance in selecting genes with heart rate effects in medaka. In previous work, it has been shown that randomly selecting 10 genes from the medaka genome resulted in only two genes affecting heart rate upon editing with CRISPR/Cas9, and both of these genes had been linked to heart function before (Hammouda et al., 2021). It should be noted, however, that our higher-than-baseline success rate stems not only from the selection of genes within association regions, but also from our broader gene prioritisation strategy, which includes the preferential validation of genes with severe loss-of-function mutations that have previously been linked to heart-related phenotypes in other species.”

5. The simulation section addressing human GWAS applicability suffers from fundamental limitations that undermine its conclusions. The simulations assume a relatively simple genetic model based on medaka QTLs, which cannot adequately reflect human complex trait architecture where hundreds to thousands of causal variants typically contribute to phenotypic variation. Critical parameters differ substantially between the medaka system and human populations, including linkage disequilibrium patterns, allele frequency distributions, and population structure. Most problematically, the simulations treat environmental factors as precisely measured continuous variables, which rarely holds in human studies where environmental exposures are typically measured with substantial error and temporal variation. Given these limitations, the conclusions about human GWAS design are not well-supported and may mislead readers. I recommend removing this entire section and focusing the manuscript on the valuable medaka-specific findings, which stand on their own merit without requiring extrapolation to human genetics.

We thank the reviewer for highlighting these important considerations. We acknowledge that our simulations do not capture the full complexity of human genetic architecture, nor do we claim to do so. We note that our simulations explicitly test environmental factor measurement error (see **Figure 6**, the E-noise rows of 1%, 10%, and 50%). We also explicitly model GWAS by proxy variants in linkage disequilibrium with the causal variant ($r^2 = 0.9$ column). However, the reviewer is correct that we necessarily had to simplify our simulations to make them computationally feasible, and we cannot account for the full complexity of confounding factors, interactions, temporal variation, or the variation in genetic and environmental trait architectures. We respectfully disagree with the suggestion to remove this section, as it is of broad interest. Nevertheless, we have performed additional supporting simulations (**Figure S10**) to demonstrate that, under our stated assumptions, the simulation framework is adequate, and we have added explicit caveats in the text.

Specifically, we state in results:

*“This simulation is, by necessity, a simplification of real-world data. To demonstrate its applicability to outbred populations, including humans, we applied the same framework to human height (**Figure S10**). The results suggest that, at least for purely genetic effects, our simulations are appropriate provided the trait of interest has effect sizes and residual variance comparable to those observed in medaka.”*

Furthermore, we have a stronger caveat section in at the start of the discussion on simulation:

“Our simulations are, by necessity, a simplification of real-world data. Although we capture certain aspects of linkage disequilibrium through tagging SNPs, we do not model interactions between QTLs in linkage disequilibrium (e.g. neighbouring loci) or interactions between population stratification and environmental factors—both of which are common sources of confounding in outbred populations. In addition, traits in

both medaka fish and humans have distinct genetic architectures and combinations of confounders, further complicated by the temporal variability of many environmental measures. Accordingly, our simulations should be regarded as best-case scenarios for the discovery of individual QTLs rather than as comprehensive models of complex trait architecture in humans. Nevertheless, they remain informative about the parameter space relevant to outbred studies."

We thank the reviewer for this constructive challenge, which prompted us both to test our simulation framework more thoroughly and to improve how we communicate its appropriateness to readers.

Reviewer #3

Review of "Discovery and characterization of gene by environment and epistatic genetic effects in a vertebrate model." In this manuscript, Welz et al. phenotype over 80 inbred medaka (*Oryzias latipes*) lines for embryonic heart rate, perform 11 sets of crosses with 8 phenotypically divergent lines, and use low-coverage WGS on F2 offspring and to identify QTLs. Commendably, the authors also experimentally validate the QTLs with gene editing (e.g., crisper) and perform simulations to determine the effect of epistasis on locus discovery in human GWAS studies. Overall, the authors performed a very large amount of well-designed work, the analyses and results are compelling, and the manuscript is very clearly written. A number of comments follow, the majority of which should be straight forward to address.

We thank the reviewer for the positive feedback on our work and address the points raised below.

Comments:

The authors identify ~ 16 QTL using methods and thresholds similar to those used in human studies where conservative thresholds are required for obvious reasons. However, 16QTL is not very many and limits the ability to account for epistatic interactions. I appreciate that there is one locus of very large effect and also appreciate the approach the authors use as re-running the analysis with that locus as a covariate. Nevertheless, I think it would be informative to use a substantially less stringent threshold(s), identify many more candidate QTLs, and re-run the GXG and GXGXE associations. This information could be largely relegated to SI, with a few additional lines to the main text summarizing what was found.

We thank the reviewer for this observation. Our significance threshold for the discovery GWAS is determined empirically as the minimum p -value observed genome-wide across 100 permutations of the model residuals, rather than being derived from human studies. In the simulation approach, however, we do use a human GWAS threshold ($p < 5 \times 10^{-8}$), as the simulations are intended to inform the design of human GWAS.

Given the complex population structure in our study and the limited sample size, we are careful to avoid false-positive effects arising from uneven allele frequencies across population subsets and other unchecked sources of confounding. High levels of linkage disequilibrium also make it very difficult to disentangle weaker QTLs from overlapping stronger loci. Therefore, results of epistatic tests among QTLs identified using a more permissive threshold could be misleading, as we cannot confidently distinguish true associations from confounding artifacts or disentangle multiple overlapping association signals.

For these reasons, we respectfully disagree with the suggestion to include an epistatic analysis across a larger set of QTLs identified under more permissive thresholds, as such an analysis would be strongly confounded by LD and thus potentially misleading.

I am a bit skeptical of imputation methods on F2 individuals. Imputation works well in highly inbred populations, but the F1 parents are highly heterozygous. I appreciate that there is a published paper documenting this method, but additional details to reduce this skepticism in future readers would be useful (were the same strains used, were any sequenced to higher depth.). Could the imputation be responsible for the fact that almost the entirety of chromosome 15 has significant p-values for QTL analyses? If not, please make it clear why.

We thank the reviewer for this observation. The samples used in this study are the same as those analysed in Pierotti *et al.*, 2024, *Bioinf, Adv.*, so the validation of the imputation process presented there fully applies here.

Imputation performs well in this population because the F2 (and F1) individuals are only a few recombination events away from the founder strains. As a result, haplotypes are largely conserved across samples, and this redundancy can be exploited for imputation. Importantly, the method does not rely on an external reference panel but instead uses internal redundancy among the sequenced samples. Such highly interrelated populations are precisely the type for which the STITCH imputation method was designed.

The fact that F1 samples are heterozygous is not critical for imputation quality by STITCH, as the method does not depend on phasing information but on conserved haplotypes, which in the F1 are identical to those in the founder inbred strains. As mentioned, extensive validation of this approach for this population is provided in Pierotti *et al.*, 2024. To clarify this connection, we revised the methods section as follows:

“The F2 medaka embryos were sequenced and their genotype was imputed with the birneylab/stitchimpute pipeline as previously described (Pierotti et al., 2024). The population analysed in that study is the same as the one used here, so readers can refer to that work for further details on the imputation process and validation.”

Regarding the reviewer’s other questions, the strains used in the referenced work are the same as those analysed here, since the same samples were used. As detailed in the current manuscript, we sequenced at high coverage one F1 individual per cross combination (10 in total) and two F2 individuals from the 72-2×55-2 cross. These 12 samples sequenced at high depth served as ground truth for validating the imputation process. For convenience, we reproduce the relevant paragraph below:

“One F1 sample per cross and 2 F2 samples from the 72-2 x 55-2 cross were sequenced at higher depth (12 samples in total, obtaining a sequencing depth of 33x to 61x). These samples were used as a ground truth for validating the imputation process.”

Further details on our validation of the imputation results are provided in Pierotti *et al.*, 2024.

Finally, we consider it very unlikely that the strong association signal observed on chromosome 15 is an artefact of imputation. Instead, it is readily explained by the high linkage disequilibrium expected in an F2 population. Moreover, as demonstrated in Pierotti *et al.*, 2024 and summarised in our manuscript, the imputation process is highly reliable in this population.

“Overall, the imputed genotypes proved to be very reliable, with an average sample-wise r^2 value of 0.996 between the genotypes called on the high coverage samples using GATK and the imputation obtained from the same samples with reads downsampled to 0.5x depth.”

Why was only a single reciprocal cross performed? Are the sex-determining loci/regions known in medaka? Is sex genetically determined? I

We thank the reviewer for this question. To assess potential parent-of-origin effects, we performed one reciprocal cross as an additional experiment. However, we did not extend this to all crosses, as investigating such effects was not the main aim of this study and would have introduced considerable logistical complexity in terms of fish husbandry. In medaka, sex is primarily determined by the DMY locus on chromosome 1 (Matsuda, 2004, *Dev. Growth Differ.*, <https://doi.org/10.1111/j.1440-169X.2003.00716.x>) although environmental factors can, in some cases, cause phenotypic sex to diverge from genetic sex (Shinomiya et al, 2004, *Zoolog. Sci.*, <https://doi.org/10.2108/zsj.21.613>).

Line comments:

Unfortunately, neither line number nor page numbers were provided in my review copy. As such the line comments follow by section and the first 3-5 words that the sentence starts with (can search to find line).

Abstract:

"showing that apparent additivity..." could this result be made more clear - "apparent additivity in human GWAS" is vague; also what are the implications for phenotypic prediction?

We thank the reviewer for this comment and edited the abstract accordingly:

"Our results suggest that the limited detection of non-additive effects in human GWAS is largely expected given the study designs and sample sizes of current datasets. Importantly, the limited contribution of interaction terms to QTL discoverability does not imply that they are irrelevant for phenotypic prediction at the individual level. Their impact on phenotypic variance — and thus on discoverability — depends not only on effect size but also on the covariance among interacting variables at the population level."

Introduction:

"Nonetheless, even in human genetics" - I thought this sentence was going to say something else entirely. Please revise the first few words for clarity.

We thank the reviewer for this observation and revised as follows:

"In fact, even in human genetics, the importance of non-additive effects for realising the goals of precision health is well recognised (Bakermans-Kranenburg & IJzendoorn, 2015; Motsinger-Reif et al., 2024) and is at the heart of the field of pharmacogenomics (Russell et al., 2021)."

"...such as dominance and epistatic effects (Currant)" - a bit vague; could specific dominance and epistatic effects be mentioned?

We thank the reviewer for this observation and revised as follows:

"Moreover, several examples of environmentally-dependent effects in humans have been reported (Bononi et al., 2020; Caspi et al., 2003; Hawn et al., 2018; Johnson et al., 2010; Modafferi et al., 2021; Polimanti et al., 2018; Rampersaud et al., 2008), as well as other forms of non-additivity such as dominance and epistatic effects. For example, Palmer et al., 2023 identified 183 phenotype–locus pairs in the UK Biobank showing genome-wide significant dominance effects, while Currant et al., 2023 reported epistatic interactions among common variants in the VSX2 and PRPH2 genes affecting photoreceptor cell layer thickness in the human eye."

The paragraph about mice is okay, but probably not needed

We thank the reviewer for this comment and understand it to refer to the third paragraph of the introduction. We consider this paragraph important because it explains our choice of medaka as a model organism, particularly in comparison with other species and considering the complex domestication history of commonly used models such as mice. In contrast, medaka strains are derived directly from the wild, providing a more unbiased representation of natural genetic variability.

"It's high fecundity, ex-utero..." - it was not immediately clear why ex-utero was an advantage - but probably is a big advantage for the phenotype of choice. Consider making more clear

We thank the reviewer for pointing out that the benefit of *ex-utero* embryonic development was not immediately clear. We have revised the text to better explain its benefit in facilitating non-invasive, scalable phenotyping.

"Its high fecundity, short generation time, small 700 Mb genome, and cost-effective husbandry make it particularly suitable for large-scale genetic studies. Moreover, its ex-utero embryonic development allows for non-invasive extraction of physiological markers in vivo due to embryo transparency, facilitating phenotyping with scalable numbers (Furutani-Seiki & Wittbrodt, 2004; Kasahara et al., 2007)."

"with recorded ecological limits (35C)" - need a reference for this statement

We thank the reviewer for this observation and referenced Yamamoto, T. 1975. Medaka (Killifish) Biology and Strains. Keigaku Publishing Company, Tokyo.

Somewhere in the legend of Figure 1, can you make it clear that the data are heuristic (i.e., not real) - I initially thought the data in panel 1 were empirical data.

We thank the reviewer for this observation and have added the following clarification to the title and legend of Figure 1:

"Schematic study design for the identification and validation of genetic variants underlying embryonic heart rate differences at changing temperature conditions."

"Note: panel 1, 2, 4, and 5 do not show real experimental data and are for illustrative purposes only"

Results:

"...we observed sub-clustering of F2..." - was the sub-clustering random or driven by cross-type? Those details should be added here.

We thank the reviewer for this question. The sub-clustering observed within the reciprocal cross is driven by the direction of the cross (72-2×139-4 or 139-4×72-2), as indicated in Figure S2 by the black and gray labels in the "Cross ID" annotation. We have revised the relevant paragraph to clarify this point:

"Within the reciprocal cross (72-2×139-4 and 139-4×72-2) we observed sub-clustering of F2 individuals according to the direction of the cross, indicating a residual degree of heterozygosity in the fish used to establish the crosses."

I really like Figure 2 - pretty amazing data! One minor comment, the double headed vs single headed arrows in panel C are confusing and not explained anywhere in the legend. Also, the reciprocal cross should be identified in the legend.

We thank the reviewer for requesting this clarification. In the figure, only double-headed arrows were used. The arrows that appeared single-headed actually represent a single arrow connecting strains 62-2 and 15-1; due to space constraints, this arrow passes beneath the box for strain 72-2. We have modified the figure to make this visually clearer:

For the reciprocal cross, we added a clarification in the legend:

“The reciprocal crosses (72-2×139-4 and 139-4×72-2) are indicated using a combined male/female symbol.”

"On the contrary,..." I did not follow this reasoning - could this be explained more clearly (also may help resolve 3rd main comment from above)?

We thank the reviewer for this question and assume they refer to the following sentence:

“On the contrary, when we looked into the marginal effect of the reciprocal cross status independent of genetics, we were able to detect significant effects on heart rate after Bonferroni correction at all the temperature treatments (21 °C: $p = 8.06e-03$; 28 °C: $p = 7.29e-08$; 35 °C: $p = 5.18e-04$).”

This indicates that we do not observe differential QTL effects depending on the reciprocal cross status, but we do detect an overall effect of the reciprocal cross. In other words, while the heart rate of individuals differs significantly between the two directions of the reciprocal cross, we cannot detect any differences in the effect of specific QTLs on heart rate depending on cross direction. To clarify this point, we added the following sentence:

“In other words, although heart rate differed significantly between the two directions of the reciprocal cross, we did not detect any cross-direction–dependent effects of specific QTLs on heart rate.”

Figure 3: Why were only 100 permutations performed? Could 1000 permutations be performed, for example.

We thank the reviewer for raising this point. We set the significance threshold as the minimum genome-wide p-value across 100 permutations of the model residuals, corresponding to an empirical significance level more stringent than 0.05 genome-wide. Specifically, the 0.05 genome-wide empirical threshold is defined as the 5th percentile of the 100 minimum genome-wide p-values obtained from these permutations, and therefore lies between the 5th and 6th smallest values. While a more stringent threshold based on 1000 permutations would be possible, we considered 100 sufficient, as this already provides a more conservative criterion than the commonly used 0.05 genome-wide threshold in GWAS.

For clarity, we note that each permutation of the residuals involves testing the full genome for associations, resulting in approximately 100×3.12 million statistical tests used to build the empirical p-value distribution.

"we observed that 21 out of 120 possible QTL pairs are significant...GxG.." - Was there any correlation between physical distance and GXG or GXGXE loci? That is, were pairs closer together in the genome more likely to be GXG or were they all on 16 separate chromosomes?

We thank the reviewer for this question. There are only a few QTLs located on the same chromosome: two QTLs on chromosome 15, two QTLs on chromosome 9, and two QTLs on chromosome 11. Among the 21 QTL pairs significant for G×G and/or G×G×E interactions, only 2 pairs are located on the same chromosome (chr15_qtl/chr15_minor_qtl and chr11_qtl/chr11_minor_qtl). Given that 3 of the 120 possible QTL pairs fall on the same chromosome, Fisher's exact test indicates that there is insufficient evidence to reject the null hypothesis that this pattern occurred by chance ($p = 0.1601$).

Figure 4 c: It would be useful to know which temperature treatments (individually) were significant after accounting for multiple comparisons. More importantly, how much is the overall significant result driven by a single or only a handful of temperature treatments?

We thank the reviewer for raising this point. We added significance labels (stars) for the individual temperature tests in Figure 4C, applying a 5% False Discovery Rate to correct for multiple testing. Editing of *ccdc141* and *sptbn1* is significant at all temperatures. For *ryr2b*, individual tests are significant above 24 °C, except at 35 °C. For *ppp3cca*, none of the individual tests are significant, likely reflecting the increased statistical power achieved by testing jointly across temperatures rather than in isolation.

"; in each of the four cases the effect validated by..." - wording is strange here - "the effect was consistent" - do you mean the predicted vs. measured shift in phenotype or just the fact that loci had an effect? Please clarify.

We thank the reviewer for this request. Our intended meaning is that, in cases where this could be assessed, the effects observed in the discovery GWAS and those observed through gene editing for the candidate genes were consistent in both direction and temperature dependence. However, for some QTLs, assessing directionality was not possible because a clear causal variant could not be identified. We have modified our sentence accordingly, as suggested:

"The gene-editing effects observed were broadly consistent in both direction and temperature dependence with the corresponding QTL effects identified in the F2 cross GWAS."

"respect to two loci- chr15_qtl and chr21_qtl..." why these two instead of those shown in Figure 5?

We thank the reviewer for this question. We used those loci because, as explained in the text, they are the only ones for which we were able to confirm wild allele frequencies. Allele frequencies are a key parameter in determining the contribution of G×G interactions to phenotypic variance. By fixing the allele frequency of one of the interacting partners, we reduce the dimensionality of the simulation space to explore.

Choice of MAF filter (and other filters) both here and for GWAS are likely important. See <https://doi.org/10.1038/s41576-024-00738-6>

We thank the reviewer for raising this point and agree that the choice of filtering strategy is critical for GWAS results. In our study, we chose not to apply MAF filters and instead relied on imputation accuracy to select reliable genetic variants (see Pierotti et al., 2024, *Bioinformatics Advances* for details). This decision is motivated by the properties of the STITCH algorithm (Davies et al., 2016, *Nat. Genet.*), which implements the Li and Stephens HMM. Variants that can be imputed with high accuracy are expected to (1) be correctly mapped to the reference genome and (2) reflect true genetic variation rather than sequencing artefacts. We therefore consider this strategy more robust than applying arbitrary thresholds based on MAF or other parameters. The original STITCH publication also describes and validates this filtering approach.

Figure 6 could be a little more clear. Rather than simple yes vs no, could you also color code and include epistasis vs no epistasis or something along those lines?

We thank the reviewer for raising this point and have modified **Figure 6** and **Figure S7** accordingly.

Figure 6 (and discussion): there is a big difference between locus discovery and phenotypic prediction; it seems epistasis would have a large effect on the latter and, as you show, a small effect on the former. This distinction could be added to the abstract and talked about more in the Discussion.

We thank the reviewer for this comment. We have revised the discussion and abstract accordingly.

Abstract:

“Importantly, the limited contribution of interaction terms to QTL discoverability does not imply they are irrelevant for phenotypic prediction at the individual level. Their impact on phenotypic variance — and thus on discoverability — depends not only on effect size but also on the covariance among interacting variables.”

Discussion:

“The same applies to epistatic effects, which we found to make only a minor contribution to QTL discoverability. Although such effects may be of large magnitude and critical for individuals carrying specific allelic combinations, their overall contribution to phenotypic variance — and thus to discoverability — likely remains small due to their combinatorial rarity.”

Discussion:

any thoughts on higher order epistatic interactions (≥ 3 QTLs/loci/genes) would be interesting to add.

We thank the reviewer for this request. While higher-order epistatic interactions are biologically interesting, they are considerably more difficult to assess due to the combinatorial explosion of possible interactions and the resulting multiple testing burden. In addition, their occurrence in natural populations is likely to be rare, as it depends on the joint frequency of three or more alleles. As suggested, we have added a paragraph to the discussion addressing this point:

“In this work, we restricted our analysis to pairwise epistasis among genome-wide significant loci. This decision was motivated by the combinatorial complexity that arises when testing a large number of loci for interactions, particularly when considering higher-order epistasis (e.g., involving three or more loci). Future studies investigating higher-order contributions may provide novel mechanistic insights but will inevitably face substantial challenges due to the multiple testing burden and the rarity of higher-order allele

combinations. An alternative strategy is to examine the effect of the same genetic variant across different medaka strains, as demonstrated in **Figure 5D**. Such comparisons provide indirect evidence of epistasis with an undefined set of background variants that differ among strains.”

great paper overall!

We thank the reviewer again for the positive evaluation of our work.

Referees' reports, second round of review

Reviewer #2:

The authors have adequately addressed most of my comments. I remain concerned about extending the model to humans without empirical support, but I leave this decision to the editor's discretion.

Authors' response to the second round of review

Reviewer #2

The authors have adequately addressed most of my comments. I remain concerned about extending the model to humans without empirical support, but I leave this decision to the editor's discretion.

We thank the reviewer for their feedback. We have addressed this comment by clarifying the limitations of our results in the “Limitations of the study” section of the Discussion as requested by the editor.